# CE-BLAST makes it possible to compute antigenic similarity for newly emerging pathogens

Tianyi Qiu[1,2], Yiyan Yang[1], Jingxuan Qiu[1], Yang Huang[2], Tianlei Xu[1,3], Han Xiao[4], Dingfeng Wu[1], Qingchen Zhang[1], Chen Zhou[1], Xiaoyan Zhang[2], Kailin Tang[1], Jianqing Xu [2] & Zhiwei Cao[1]

Major challenges in vaccine development include rapidly selecting or designing immunogens for raising cross-protective immunity against different intra- or inter-subtypic pathogens, especially for the newly emerging varieties. Here we propose a computational method, Conformational Epitope (CE)-BLAST, for calculating the antigenic similarity among different pathogens with stable and high performance, which is independent of the prior binding-assay information, unlike the currently available models that heavily rely on the historical experimental data. Tool validation incorporates influenza-related experimental data sufficient for stability and reliability determination. Application to dengue-related data demonstrates high harmonization between the computed clusters and the experimental serological data, undetectable by classical grouping. CE-BLAST identifies the potential cross-reactive epitope between the recent zika pathogen and the dengue virus, precisely corroborated by experimental data. The high performance of the pathogens without the experimental binding data suggests the potential utility of CE-BLAST to rapidly design cross-protective vaccines or promptly determine the efficacy of the currently marketed vaccine against emerging pathogens, which are the critical factors for containing emerging disease outbreaks.

[1] Shanghai 10th People's Hospital, School of Life Sciences and Technology, Tongji University, Shanghai 200092, China. [2] Shanghai Public Health Clinical Center & Institutes of Biomedical Sciences, Shanghai Medical School, Fudan University, Shanghai 200032, China. [3] Department of Mathematics and Computational Science, Emory University, Atlanta GA, USA. [4] Department of Computer Science, University of Helsinki, Helsinki FI-00014, Finland. These authors contributed equally: Tianyi Qiu, Yiyan Yang, Jingxuan Qiu. Correspondence and requests for materials should be addressed to J.X. (email: xujianqing@shphc.org.cn) or to Z.C. (email: zwcao@tongji.edu.cn)

Emerging and re-emerging diseases caused by infectious pathogens are identified almost every year and remain a continuous threat to public health. Recent examples include influenza, avian flu, dengue, severe acute respiratory syndrome (SARS), and Ebola hemorrhagic fever (EHF), with the latest being microcephaly caused by the zika virus[1]. To combat these epidemics, vaccines are consistently needed for the purpose of disease control and prevention. A critical step in vaccine development is to characterize the antigenicity difference among various pathogens so as to select or design proper immunogens that are able to raise cross-protective immunity. To date, determining antigenic variation of the emerging pathogens has relied heavily on the results from the immune-binding assays. For instance, the hemagglutination inhibition (HI) assay is traditionally performed to determine the antigenic changes in circulating influenza viruses from those of the previous vaccines[2]. Antisera from multiple donors are routinely screened for binding against virus strains in search of a potential broad-spectrum antibody for human immunodeficiency virus (HIV)[3–5]. Recently, comprehensive serological tests were accomplished on both animals and vaccinated or infected humans to calibrate the serological relationships between the subtypic dengue viruses (DENV)[6]. Despite the wide adoption for common infectious diseases, immune-based experiments are often found to exhibit limited application in the case of significant outbreaks or emergence of new virus subtypes, owing to various factors of mobility, antiserum dilution, standardization, and automation. Thus, new automated technologies with high-throughput and quick response are always desired, so as to meet the increasing demand of newly emerging epidemics. Accordingly, the development of computational strategies independent of immunoassays may be helpful for assisting the antigenicity measurement in a rapid and timely manner.

Till now, in silico methods to compute antigenicity have been developed primarily for only a few specific pathogens, based on the knowledge acquired from massive accumulation of the historical experimental data, such as for influenza virus or foot-and-mouth disease virus (FMDV)[7–9]. However, for numerous other pathogens and new pathogens for which the binding assays remain sparse or insufficient, no computational model has yet been reported. In this study, we designed a generalized and immunoassay-independent tool, Conformational Epitope (CE)-BLAST, to predict the antigenicity of different pathogens.

Similar to the concept underlying sequence BLAST with the sequence similarity inferring functional similarity, CE-BLAST aims to compare the conformational epitopes directly to suggest the relative antigenicity distance between antigens. In the adaptive humoral immune system, the pathogenic antigens will be recognized and bound by specific antibodies at the conformational epitopes generally comprising several segments that are discontinuous in sequence, but close in three-dimensional (3D) conformation[10,11]. This recognition process features high sensitivity and specificity, where only the mutated antigens with highly similar conformational epitopes are able to cross-react with the same antibody. Arising mutants with substantially different conformational epitopes are likely the causing antigenic variants to previous vaccines, and may lead to new outbreaks in the community[12–14]. Therefore, comparing the conformational epitopes directly may provide clues to infer antigenic similarity of the pathogenic antigens. This algorithm takes complete consideration of the structural and the physicochemical microenvironment variations from a 3D viewpoint caused by mutations, which are summarized into a comprehensive fingerprint for each epitope residue. For each input antigen with the structure information, the conformational epitope will be translated into a series of fingerprints and compared with the predefined or user-uploaded datasets through CE-BLAST, then a list of hit-epitope structures with predicted similarity scores will be provided in descending order as output.

The ability of CE-BLAST to detect the antigenic variance is rigorously evaluated using different sets of immune-assay data on both intra- and inter-subtypic pathogens, as well as cross-virus cases. It is initially tested with intra-subtypic pathogen data of influenza A/H3N2 antigen including 3867 historical HI assays, and then combined with the experimental validation on a new antigen of A/H3N2. Then, its ability to classify serological relationships is further confirmed on DENV subtypes via 1072 serological data results. Notably, the application scope of CE-BLAST is extended to a cross-virus case to suggest the potential cross-reactive epitopes between ZIKV and DENV in the Flavivirus family. For convenient use, a web server has been constructed with built-in epitope libraries containing simulated structure databases of the HA antigen for influenza virus (A/H1N1 & A/H3N2), Envelope (E) antigen for DENV and ZIKV, and known conformational epitopes derived from the Protein Databank (PDB) immune complex. The web server of CE-BLAST can be accessed at http://badd.tongji.edu.cn/ce_blast/ or http://bidd2.nus.edu.sg/czw/ce_blast/.

## Results

**Model construction of CE-BLAST.** The design of the CE-BLAST model encompasses three steps: (1) deriving a group of fingerprints for each conformational epitope, (2) aligning the conformational epitopes according to their fingerprints, and (3) scoring the similarity according to the epitope alignment. In the first step, the epitope fingerprints are composed of individual fingerprints of each epitope residue, which are described by the residual layout and the physicochemical properties of the residual microenvironment via spin-image and shell-structure models. In the second step, a "seed-grow" strategy is subsequently adopted to identify the best local alignment, according to the fingerprint comparison between conformational epitopes. In the third step, the similarity score considers not only the number of matched residues, but also the evolutional distance between matched positions, as well as the similarity of the corresponding microenvironments for each residue. CE-BLAST begins with conformational epitope structures and requires no experimental binding data. Such unsupervised performance ensures its adaptability for new antigens without prior assays. Additional details can be found in the Methods.

The workflow of CE-BLAST is illustrated in Fig.1. The algorithm accepts epitope structures in the protein data bank (PDB) format as input and then converts the structure information into epitope fingerprint. Users can search against the built-in epitope database or search within their input data files to find antigenically similar epitopes for the queried files. Finally, the results are provided as a hit list including the ID of each hit epitope and the corresponding similarity score.

In view of the extensive computational time required for fingerprints derivation, we have modeled thousands of representative HA structures for influenza H3N2 and H1N1 antigens, and calculated their epitope fingerprints based on the predefined epitope sites. Furthermore, the E protein of two Flaviviruses (both monomer and dimer), DENV and ZIKV, were also premodeled and added into the built-in database. Currently, CE-BLAST contains three built-in epitope databases including: (1) 559 known epitope structures derived from the immune complexes in the PDB database; (2) conformational structures of 1284- and 1725-modeled HA structures representing 16,672 H1N1 strains and 15,238 H3N2 strains, respectively; and (3) conformational structures of 1143- and 68-modeled E protein representing 4081

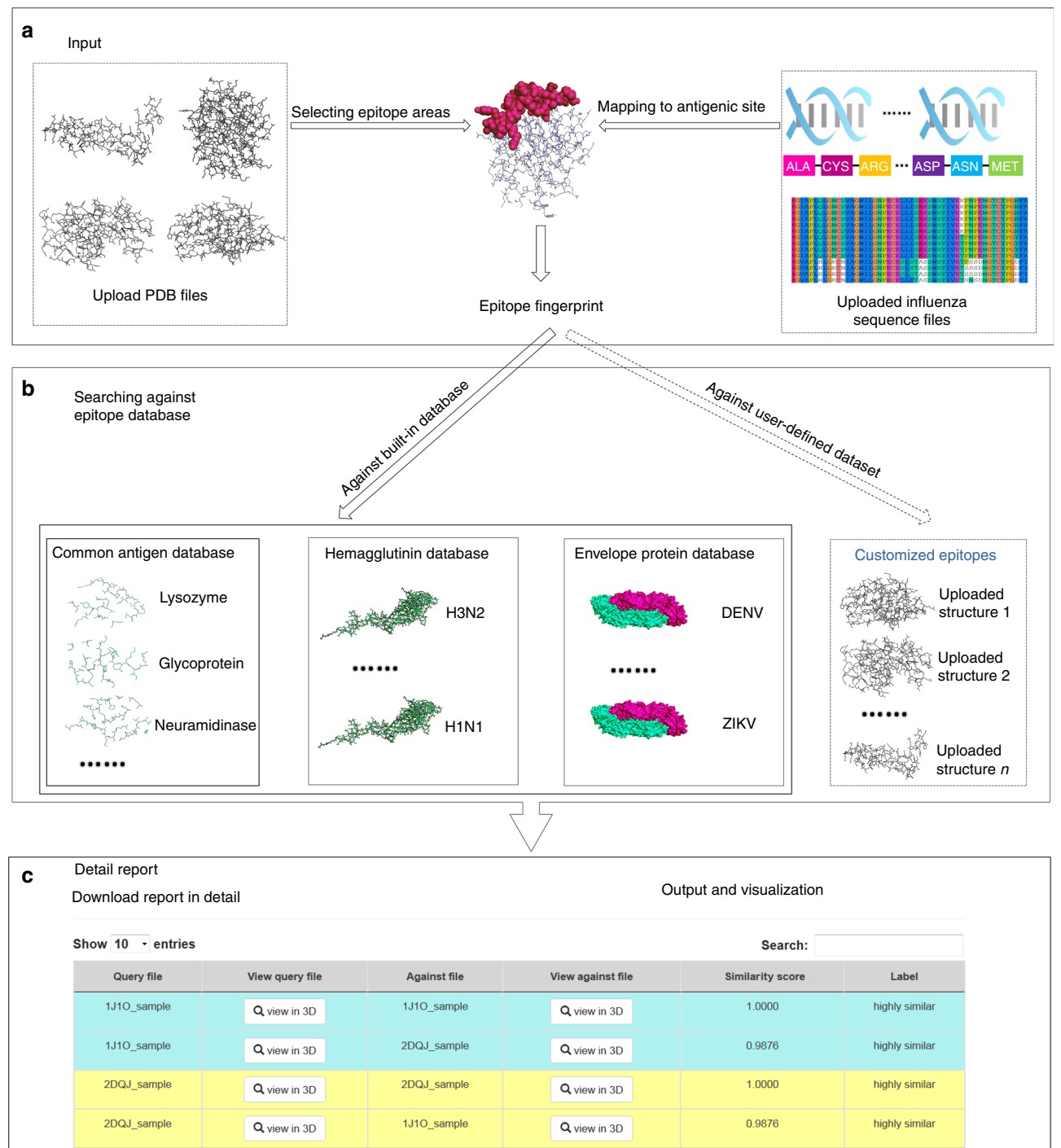

**Fig. 1** Model workflow of CE-BLAST. **a** The input files for CE-BLAST can be either the PDB structure of any protein antigen or the HA sequences of influenza A/H1N1 and H3N2 antigens. After the epitope sites are selected, CE-BLAST can automatically calculate the fingerprints for each epitope structure. **b** The epitope fingerprints are used to search against a built-in epitope database or a self-defined dataset that is uploaded by the user. **c** Output results are provided as a list of hit epitope structures with similarity scores in descending order. The user can also compare the structural differences using visualization links

DENV strains and 441 ZIKV strains, respectively. Validations of CE-BLAST on conformational epitopes can be found in Supplementary Note 1.

**High and robust performance on HI data of influenza H3.** To test whether CE-BLAST can predict the cross-reactivity of intra-subtypic pathogens, influenza H3 was initially selected owing to

the massive accumulation of HI assay values and sequence data. In this study, a complete historical HI dataset was collected to validate CE-BLAST, as well as to test the performance of the available tools specific to influenza[15–17]. Mutual HI assay values of 3867 HA pairs of influenza A/H3N2 strain were collected, representing the most abundant HI validation dataset yet reported (Supplementary Note 2). The antigenically similar or varied HA pairs were then classified according to the classical cutoff of

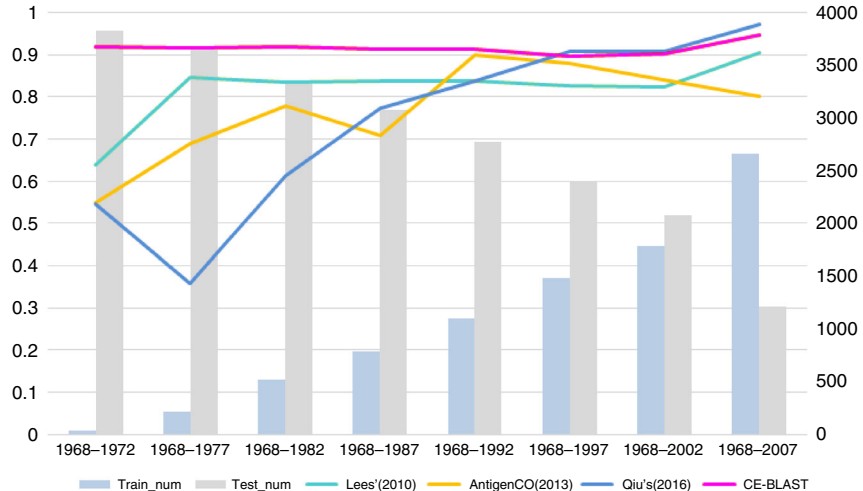

**Fig. 2** Performance comparison between CE-BLAST and peers on mutual HI data of 3867 HA pairs of influenza H3. The *X* axis represents different simulation time points with an increasing window of 5 years. Blue bars show the numbers of training data within the time period, whereas gray bars represent those remaining as testing data, with the corresponding values indicated on the right. Each colored line shows the performance (AUC value) of the computational model, corresponding to the value on the left

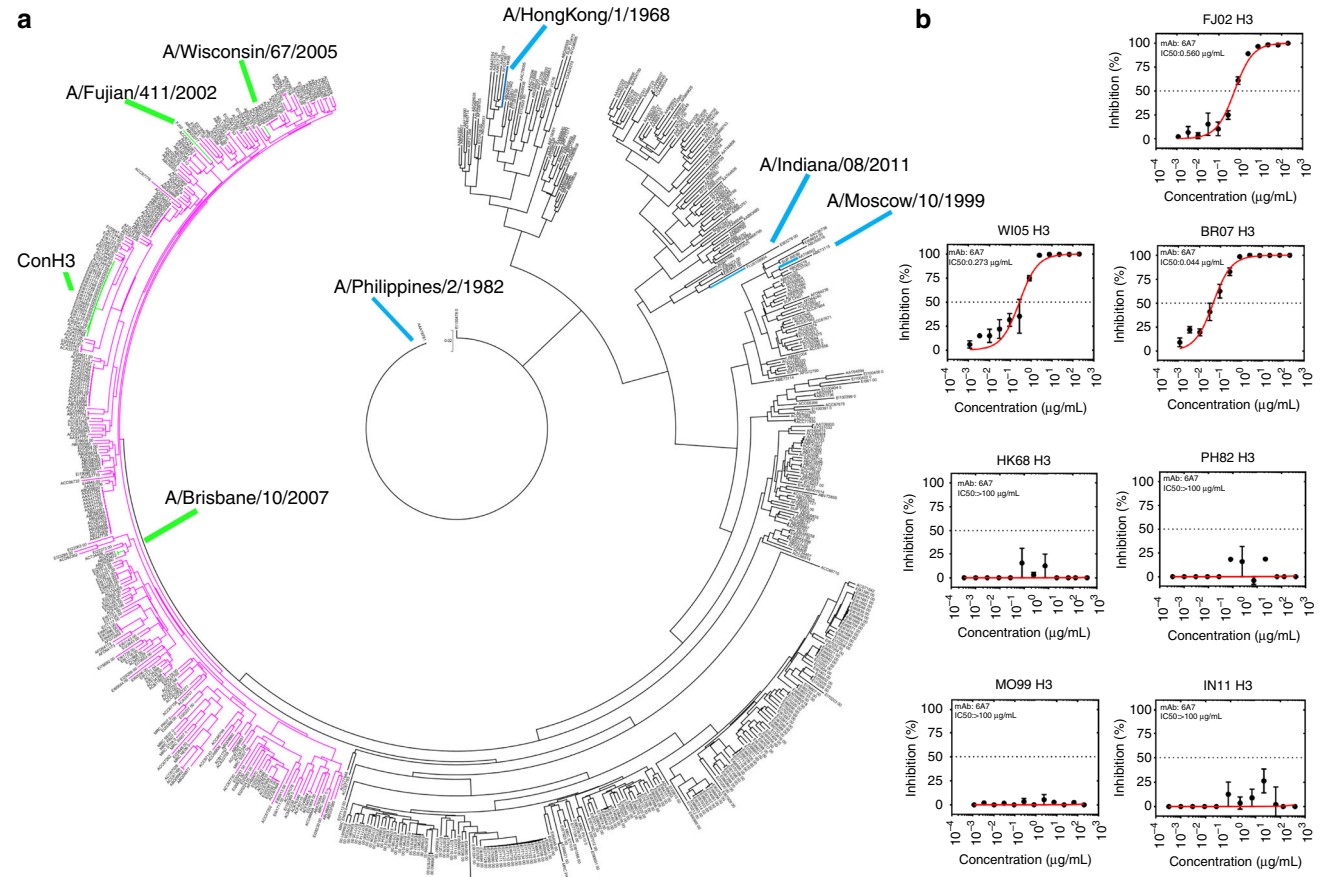

**Fig. 3** Predicting the protective spectrum for a new vaccine (Con H3) of the influenza A/H3N2 strain by CE-BLAST. **a** Antigenic clustering results between HA epitopes of 679 influenza strains. Strains with identical HA epitopes as the new vaccine were marked in green and labeled as Con H3. The pink region shows the potential antigenically similar or cross-reactive strains to Con H3. The locations of the three strains inside the spectrum are marked in green, whereas the other four strains outside the spectrum are marked in blue. **b** Inhibition concentration for the seven tested strains with the monoclonal neutralization antibody derived from Con H3-immuned mice. IC 50 values were calculated by fitting

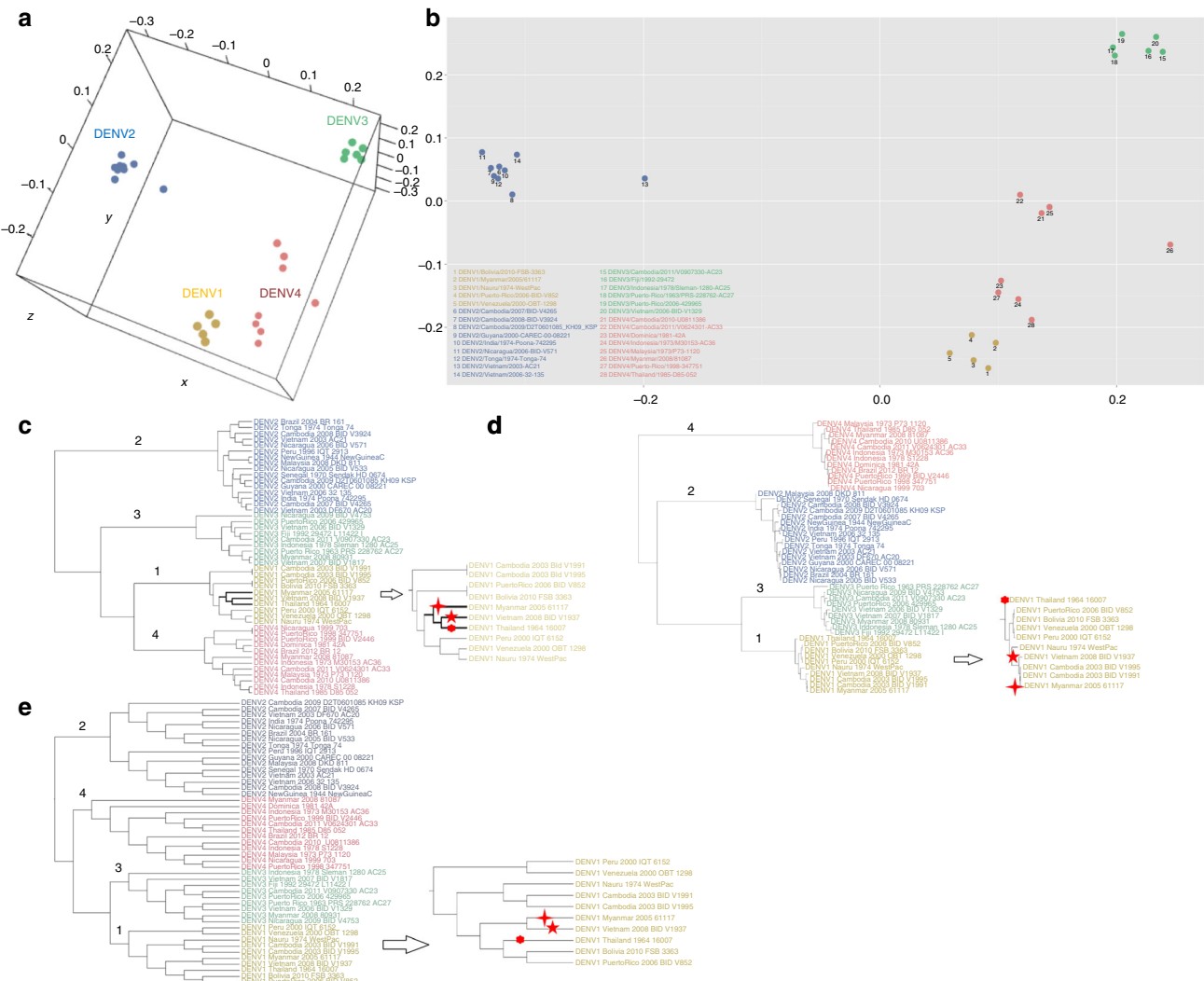

**Fig. 4** Subtype grouping of dengue virus by CE-BLAST. **a** 3D antigenic mapping of 28 dengue virus strains based on the serological data from Katzelnick et al.[6] by MDS. **b** 2D antigenic mapping of Fig. 4a. **c** Antigenic clustering of 47 strains by CE-BLAST similarity score. **d** Traditional grouping by sequence phylogenetic tree of 47 E protein sequences. **e** Traditional grouping by structural clustering tree of 47 E proteins based on RMSD scores of the Multiprot[19]. Strains DENV1/Vietnam/2008-BID-V1937, DENV1/Thailand/1964/16007, and DENV1/Myanmar/2005/61117 were marked with star, dot, and cross, respectively in **c**–**e**

the antigenic distance $(D_{ab})$ transformed from mutual HI values[17] (Supplementary Note 3). The potential cross-reactivities of the corresponding HA pairs were also predicted by CE-BLAST as similar or varied after structural modeling (Methods) of the 679 HA1 antigens. Compared to the results from HI tests, a high classification performance with area under ROC curve (AUC) value over 0.917 could be achieved by CE-BLAST on 16 classical antigenic sites. Results for different antigenic sites were also tested with similar performance (Supplementary Fig. 1).

In addition, we simulated the prediction results of CE-BLAST and other peers by different training data size with different data from 1968 to 2013 via a sliding window of 5 years, with training data continually increasing and testing continually decreasing (Methods). As almost all the available in silico tools of influenza comprise supervised models, three assay-trained methods were chosen as representative peers including Lees' method[17], AntigenCO[15], and a most recent method from Qiu[16], considering their repeatability and accessibility. As shown in Fig. 2, the overall prediction abilities of the supervised models varied differently across different dates, with AUC value below 0.65 at the beginning of testing period in 1972. When the size of training

data kept increasing, their performances become relatively stable with AUC value over 0.8 after 1992. In comparison, CE-BLAST gave high performance of AUC value of around 0.9 from the beginning of 1972, and maintained a consistently high and stable AUC value across the entire testing periods. As an unsupervised method, the prominent value of CE-BLAST appears to be able to process the fast antigenic matching of new antigens where no appropriate serology exists. Subsequently, we thus extended the prediction ability of our tool to new antigens from different pathogens.

**Reliable prediction for a new influenza H3 vaccine**. A new vaccine, named "Con H3", was artificially designed for influenza H3 without HI data through the consensus sequence of the reported A/H3N2 strains from the years ranging from 2006 to 2009, obtained from the National Center for Biotechnology Information (NCBI). After querying against the 679 HA epitopes described above, CE-BLAST gave a potential similarity profile, with the protection spectrum of Con H3 shown in pink in Fig. 3a. Then, seven strains (Supplementary Table 1) including three inside and four outside of the predicted protection spectrum were

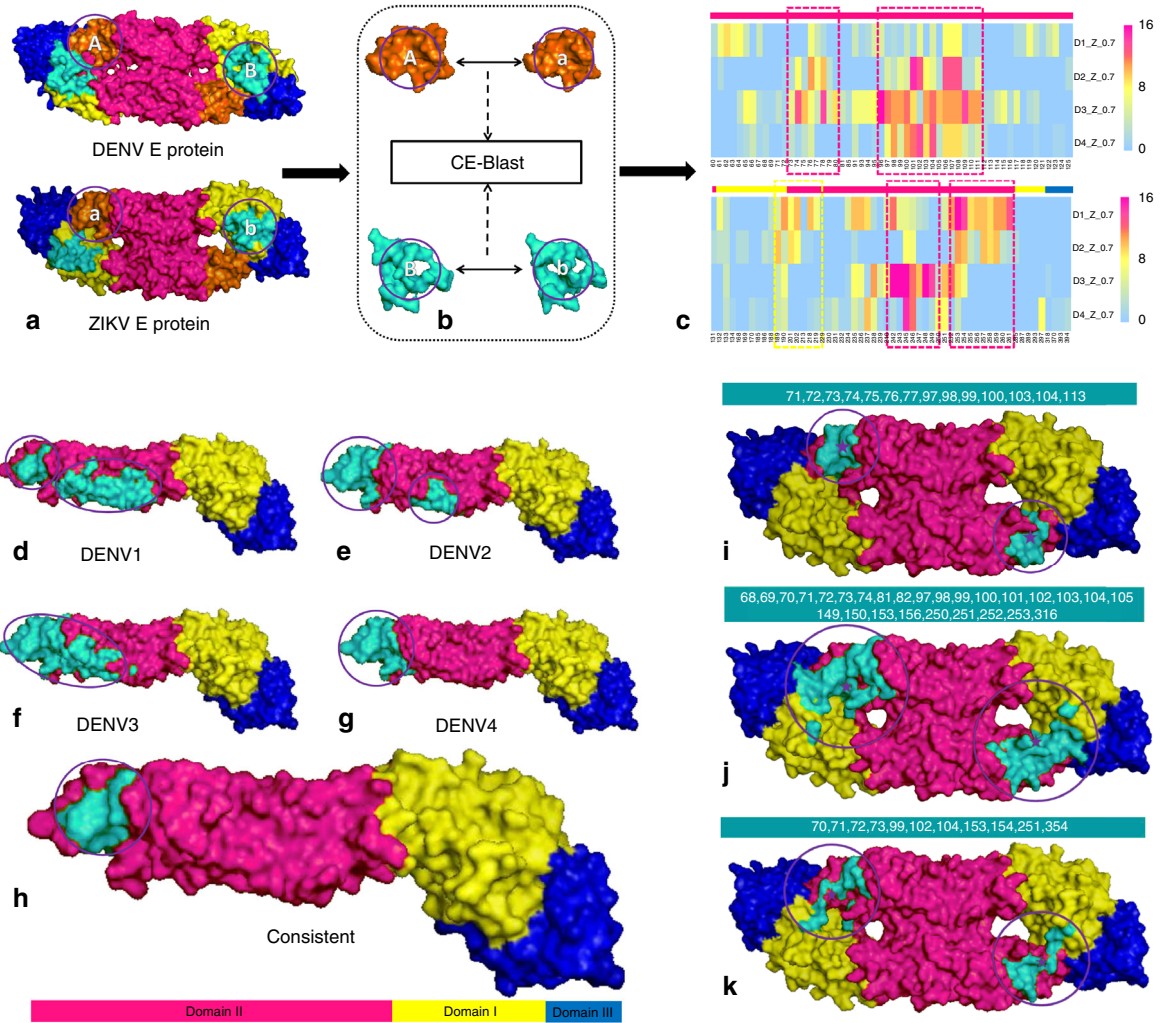

**Fig. 5** Predicting the potential cross-reactive epitope between DENV and ZIKV. **a–c** Workflow of the potential cross-reactive area (CRA) detection by CE-BLAST between ZIK and DENV. **a**: Two E antigens to be compared with domains I, II, and III marked in yellow, magenta, and blue, respectively; **b**: circular patches are screened and compared on the antigen surface; **c**: the cross-reactive frequency among sampling structures between corresponding patches predicted by CE-BLAST. Each patch is labeled by the center residue in the column, and each row represents four DENV types. Magenta dashed boxes show the consistent CRAs across different DENV subtypes, whereas yellow box shows the weak one. Residues in different domains are marked accordingly on the bars over the heat map. **d–h** Potential cross-reactive epitope (CRE) mapping to the E monomer structure of ZIKV. **d–g**: the predicted CRE is shown in turquoise for four DENV serotypes respectively; **h**: overlapping CRE of ZIKV across DENV subtypes. **i–k** Predicted CRE of the E dimer structure of ZIKV, compared with experimental results. **i**: predicted CRE by CE-BLAST for the E dimer; **j**: binding interface derived from the crystal structures (PDB id:5LCV); **k**: important residues computed by interaction force from Barba-Spaeth et al.[22]. All CREs have been circled for clarity

randomly selected for experimental validation. Sequence comparison between "Con H3" and the seven selected strains can be found in Supplementary Fig. 2. Notably, all the Con H3-immunized mice (5 mice per group) mounted significant neutralization activities against the three strains inside the protection spectrum, yielding geometric mean titer (GMT) values of 12,150 against WI05 H3 (A/Wisconsin/67/2005), 2786 against BR07 H3 (A/Brisbane/10/2007), and 2111 against FJ02 H3 (A/Fujian/2002). In contrast, only marginal responses (≤400) were elicited against MO99 H3 (A/Moscow/10/1999). No responses to the other three strains (≤50) were observed in comparison with the control group (Supplementary Table 2).

To further corroborate the above neutralization data, we generated monoclonal neutralizing antibodies from Con H3-vaccinated mice. A cross-reactive neutralizing antibody designated as 6A7 was identified from among 2400 fusion-cell clones between H3-immunized mice splenocytes and mice myeloma cells, with neutralization activities tested subsequently. The

experimental results, shown in Fig. 3b, agree well with the CE-BLAST predictions (Supplementary Table 3). In addition, the prediction accuracy of the three chosen peer algorithms was also tested on the new vaccine of "Con H3" at different time points, the results from which demonstrated that CE-BLAST out-performed the three peers, in terms of both accuracy and reliability (Supplementary Tables 4-6).

**Correct prediction of serological topology for DENV subtypes.** To further test the generality of CE-BLAST toward newly emerged pathogens, the antisera data of DENV were collected from a large-scale study on the African green monkey[6]. In this cited study, 36 sera samples derived from the monkeys injected with corresponding vaccine strains were tested individually against 47 DENV strains of four different serotypes. After removing the un-interpretable data with undone and self-reactive titers <10, the remaining titer data of 1072 strain pairs were

included as our validation set. We modeled 47 E protein structures for CE-BLAST according to the sequences provided in the paper[6]. Unlike for influenza, no empirical titer threshold has been reported as being able to classify the antigenic similarity or variance for DENV cases. According to the statistics of the available data[6], the titer value for over 90% of the self-reactive pairs was over 20. Thus, three different values of 15, 20, and 40 were tentatively chosen as classification thresholds for further testing. Accordingly, the classification results of CE-BLAST achieved AUC values of 0.857, 0.894, and 0.899, respectively, of the 1072 strain pairs.

Next, the antigenic grouping results of CE-BLAST were compared with that from the classical sequence similarity and structure similarity, based on experimental serological topology. Figure 4a, b shows the serological topology of experimental grouping between DENV strains by the multidimensional scaling (MDS) method[18] after data cleaning and normalization (Methods). It could be observed that serotype 1 clusters closely with serotype 4, whereas serotype 2 clusters the farthest from the remainder. In Fig. 4c of the CE-BLAST results, the four serotypes of DENV could be correctly predicted and clustered. In comparison to grouping topology, serotype 1 was first clustered with serotype 4, whereas serotype 2 is the farthest from the remaining strains, which completely matches with the experimental topology (Fig. 4a, b). However, in sequence-based clustering of Fig. 4d, serotype 1 was first clustered with serotype 3, followed by serotype 2, and last with serotype 4, which disagrees with the experimental results. Neither could the structure method achieve the correct topology for DENV subtypes, as displayed in Fig. 4e.

In addition to inter-subtypic DENV, CE-BLAST also yields better prediction for intra-subtypic pathogens than sequence-based or structure-based methods. Taking serotype 1 as an example, strain DENV1/Vietnam/2008-BID-V1937 (DENV1-V1937, marked with a star) in Fig. 4c was first clustered with DENV1/Thailand/1964/16007 (DENV1-T16007, marked with a dot) and then with DENV1/Myanmar/2005/61117 (DENV-M61117, marked with a cross), followed by others. This topological structure indicated that the antigenicity of DENV1-V1937 was closest to that of DENV1-T16007, followed by DENV1-M61117. Our prediction is strongly corroborated by experimental results from either 1 month or 3 month post-infection sera. The titer value of DENV1-V1937 vs. DENV1-T16007 is the highest among all pairs between DENV1-V1937 and the 47 tested strains, followed by DENV1-M61117 with the second highest titer values, indicating the close antigenicity between DENV1-V1937 and DENV1-T16007 (Supplementary Data 1). In contrast, neither the sequence-based nor the structure-based methods could suggest the best serological relationship within DENV subtypes, as being displayed in Fig. 4d, e. Therefore, the CE-BLAST model appears to give the best inference of serological similarity for DENV subtypes, compared to the classical sequence- or structure-based methods.

**Capturing the cross-reactive epitopes between DENV and ZIKV.** The obtained results indicted the unique ability of CE-BLAST to predict the antigenic similarity for intra- and inter-subtypes of new pathogens. Next, CE-BLAST was tested across different viruses to detect the potential cross-reactivity between the latest arising pathogen of ZIKV and the available pathogen of DENV. ZIKV is a member of the Flavivirus family, which recently emerged from Brazil and quickly became a significant public health concern. Evidence shows that ZIKV infections may lead to neurological complications such as Guillain–Barré syndrome in adults[20] and micocephaly in newborns[1]. Several reports

discovered that the antibodies isolated from patients with dengue had the potential to cross-react with ZIKV[21,22]. As the main target of neutralizing antibodies, the E protein was reported to share high structural similarity with a root-mean-square deviation (RMSD) of 1.1 Å and overall sequence identity of 53.9% between ZIKV and DENV[23].

To predict the potential cross-reactive epitopes between ZIKV and different DENV subtypes, four representative E proteins were randomly sampled from our dataset for each DENV subtypes, as well as for ZIKV. For the convenience of computer screening, round patches of the viruses were collected for each residue on the protein surface after structure modeling of E monomers and dimers, respectively (Methods). Then, the surface patches of ZIKV were compared with the corresponding patches in DENV subtypes through CE-BLAST. Potential cross-reactive patches (CRPs) were marked when their similarity scores rose above a certain threshold. The frequency of CRP labeled by the center residue was mapped onto a heat map resulting from binary comparison between 4 ZIKV and 4 DENV structures, as shown in Fig. 5a–c, with different rows representing DENV subtypes. Additional results can be found in Supplementary Figs. 3 and 4. Although the in silico cross-reactivity frequency could vary among different subtypes of DENV, strongly consistent CRPs in domain II and additionally weak CRPs in domain I could be detected across DENV subtypes (Fig. 5c). Our prediction is supported by the experimental results from a study by Stettler et al. on testing the reactivity ability of domain I/II and domain III in the E protein monomer[22].

It is noted that the true epitope is often irregularly shaped, whereas the above circular surface patch is artificially over-simplified for convenient screening purposes. The same cross-reactive epitope (CRE) residue may be contained by different artificial patches, and particular artificial patches covering sufficient CRE residues are more likely to constitute the CRPs. Thus, the overlapping of such CRPs likely indicates the location of true epitopes. Subsequently, individual residues in each in silico CRP of a given subtype are first mapped onto the 3D surface of the E antigen. The concentrated areas above the average are shown in turquoise in Fig. 5d–g, hinting at the potential CRE between ZIKV and DENV subtypes, respectively. Subsequent overlapping of subtypic CREs suggested the turquoise region in Fig. 5h as the potential CRE of ZIKA virus across DENV subtypes

A similar strategy was applied to E protein dimer structures. The computed CRE across DENV subtypes was strongly hinted in domain II, albeit only slightly in domains I and III from the opposite chain in the dimer structure (Supplementary Fig. 5). The entire CRE predicted for the E dimer involves 14 surface residues, as labeled in Fig. 5i. Notably, the computed CRE is highly overlapping with results from the crystallization work of immune-complexes by Barba-Spaeth et al.[21]. In particular, 71% of our computed CRA residues are located in the binding interface derived from the structure complex between the antibody and the E protein dimer (Fig. 5j), with 45% of the important residues suggested by Barba-Spaeth et al. included in our prediction (Fig. 5k).

## Discussion

Predicting the cross reactivity for new pathogens is highly challenging, particularly when the experimental data is still insufficient. In this study, we designed a unique model to achieve such prediction by comparing the conformational epitopes of different antigens. Comprehensive validations confirmed the high and stable performance of CE-BLAST. The explanation for the success of CE-BLAST in sensitively detecting the antigenic change lies in

the design of the algorithm. Firstly, the 3D residual layout difference in the whole antigen structure caused by mutations can be recorded by different rotating planes of spin-images of each epitope residue. Similarly, the physicochemical change of the microenvironment caused by the mutation in whole antigen can also be described through shell models of each epitope residue. Thus, for a pair of antigen structures with only one residual difference in the surface epitope, the coordinate displacement derived from classical structure alignment might mainly focus on the local mutated site, and therefore is usually minor. Conversely, in the CE-BLAST model, the fingerprints of all the non-mutated epitope residues will also change accordingly. Coupled with the substantial penalty from the BLOSUM matrix, the sensitivity to measure the overall difference is thereby largely increased in CE-BLAST.

Furthermore, our model enables local search of the most similar subareas between epitopes. As our similarity score is normalized by self-size, the score of epitope A querying against B may be different from that of B against A, if they have different sizes. Therefore, we tested reciprocally in the case of ZIKV and DENV. A general workflow was also proposed for cross virus reactivation by modeling the representative antigen structures. Surface patches are artificially rendered as circles for simple calculation when the shape of the real epitope is totally unknown. Highly cross-reactive patches often suggest the inclusion of more cross-reactive epitope residues. After overlapping the individual residues from cross-reactive patches, the subtype-specific and subtype-common CREs can be suggested.

Despite the generality, we also found the limitation of our models. As CE-BLAST calculates the similarity based on antigen structures, incomplete structures will reduce its performance. In addition, heavy post-translational modified structures may also influence the accuracy, such as in the case of the gp120 antigen of HIV. To summarize, we designed a new algorithm for the possible inference of antigenicity similarity, particularly for newly emerged pathogens. CE-BLAST may potentially be useful for the following applications: (1) inferring the relative antigenic distance between the mutated antigens, (2) predicting serological classification for pathogen subtypes, and (3) suggesting the potential cross-reactivity across viruses. Subsequent improvements will be further elaborated on post-translational modification (PTM) antigens, parallel computing, and refined models that are tailor-made for specific proteins.

## Methods

**Data source**. Hemagglutinin data of influenza viruses: HA1 sequences were collected from international databases and reports (Supplementary Note 2). A total of 14,891 HA1 sequences longer than 327 amino acids were retained for influenza A/H3N2 and 16,672 HA1 sequences longer than 325 amino acids were retained for A/H1N1. Based on a sequence identity of 99%, 1725 and 1284 unique HA clusters were formed for A/H3N2 and A/H1N1, respectively. Representative structures were built for each cluster with randomly selected sequences within the cluster via homology modeling (Modeller 9.11)[24].

The HI assay values were obtained for influenza A/H3N2 from the reports of international organizations along with published papers (Supplementary Note 2). For strain $a$ and strain $b$ HA sequences, the antigenic distance $(D_{ab})$ was calculated only when the four individual HI values ($H_{aa}$, $H_{ab}$, $H_{bb}$, $H_{ba}$) were available. In this way, 3867 $D_{ab}$ values for non-redundant HA pairs were derived from 288 unique HA sequences, covering 3539 strain pairs from 1968 to 2013, as different HA sequences were found under the same strain name. The dominant classification of each pairs could be detected following the protocol in Supplementary Note 3. Among the non-redundant HA pairs, 2286 were experimentally confirmed as immune-escaping according to HI results, whereas 1581 were defined as antigenically similar.

We then split the data according to different time periods to simulate the prediction ability of each algorithm. In each simulated year ($X$), the selected supervised methods were trained by data collected from 1968 to $X$; and the remainder were used to testing data to evaluate the prediction ability. In this study, eight different time periods were selected ($X = 1972, 1977, 1982, 1987, 1992, 1997, 2002,$ and 2007) and tested respectively.

Envelope protein data of dengue and zika viruses: E protein sequences were collected from the virus variation database of NCBI, with host set as human. A total of 4081 E protein sequences longer than 493 amino acids were retained for DENV 1–4 and 441 E protein sequences longer than 505 amino acids were retained for ZIKV. Based on a sequence identity of 100%, 1143 and 68 unique E protein clusters were formed for DENV and ZIKV, respectively. Representative structures were built for each cluster with randomly selected sequences within the cluster via homology modeling (Modeller 9.11)[24].

Epitope structure: A total of 421 PDB IDs that included 559 epitope structures were identified from the PDB[25] database, with key words including antibody, antigen, Fab, Fv, Fc, IgG, and immu*, with a resolution better than 3.0 Å and with a protein antigen length of more than 50 residues. For each PDB complex, epitope residues were determined by the nearest atom distance to antibody residues (≤4.0 Å). Finally, 559 epitope structures were defined as known conformational epitopes database in CE-BLAST.

**Algorithms**. To align the two epitope structures, CE-BLAST first identifies the seed residue pairs in different subareas between the two structures. Subsequently, the alignment starts from the seed and gradually extends to the neighboring area to match the similar residue pairs. Then, the similarity score can be calculated according to the aligned epitopes. For each pair of queried epitope (A) and target epitope (B), the overall steps of the algorithm will be:

Step 1: Identify the seed pairs between A and B via epitope fingerprints:

Use the spin-image system to generate structural fingerprints for each epitope residue in epitopes A and B (see Structural fingerprint generation via the "spin-image" system);

Add the physicochemical fingerprint (see Physicochemical fingerprint in the shell layers) to the structural fingerprint for each epitope residue.

Identify the seed pairs based on the epitope fingerprints.

Step 2: Use the "seed grow" strategy to find the most similar subclusters in epitopes A and B (see Epitope alignment based on the "Seed grow" strategy).

Step 3: Calculate the overall similarity for the epitopes (see Similarity score of the aligned epitopes).

**Identifying the seed pairs**. The seed residue pairs are those in the similar neighboring environments, in terms of both residue layout and physicochemical properties. A set of fingerprints was designed to describe the local environment for each residue in an epitope.

Structural fingerprint generation via the "spin-image" system: The spin-image system was initially designed to represent 3D objects for efficiently solving the object recognition and reconstruction problems. The spin-image system aims to project the neighboring residue layout to a two-dimensional (2D) array by rotating the dynamic plane of each epitope residue[26]. An input epitope structure will be described by the collection of 2D images, defined as spin-images. Each epitope residue is finally recorded as a unique image of a 2D array describing the local residue layout around the target residue. In this manner, an epitope surface can be represented by a group of spin images.

Each epitope residue $r_i$ is simplified as a point $P_i$ by its alpha carbon atom, C$\alpha$. Then, the geometric center C of the whole epitope is calculated by averaging the 3D coordinates of all the epitope residues. The center C is set as the origin of the coordinate system. The vector $\overrightarrow{CP}$ is set as the rotating axis of the dynamic plane. Along with a fixed-size rotating plane rotating around $\overrightarrow{CP}$, all the surface residues in an epitope can be projected onto a certain position in the plane. This plane can be divided into a 2D grid by appropriate horizontal and vertical pixels (Supplementary Fig. 6). Different plane sizes and grid resolutions were tested. An optimized plane size and grid resolution were selected (Supplementary Fig. 7 and Supplementary Note 4).

Physicochemical fingerprint in the shell layers: As the physicochemical properties of hydrophobic interactions, hydrogen-bond and electrostatic interactions were reported to play essential roles in the specific binding of an antigen and antibody[27,28], we presented a shell model to add these physicochemical properties at different layers around the target residue P. By means of the shell structure, the hydrophobicity, hydrogen bonding, and electrostatic interactions[29] of the neighboring residues were summarized in the shell layers according to the distance of the neighboring residue and the target residue P (Supplementary Fig. 8). After optimization, layers of shells within 20 Å of P were generated at a step size of 2 Å.

Finally, seeding residue pairs could be identified between two epitope surfaces based on the residue fingerprints of the structural environment and physicochemical properties.

**Epitope alignment based on the "seed row" strategy**. Step 1: Pearson correlation coefficients are calculated between each residue pair between epitope A and B, based on the residue fingerprints. The most similar residue pair with the highest Pearson coefficient will be taken as "seed 1".

Step 2: Within a defined distance to "seed 1" (seeding distance), all the neighboring residues are compared between the two epitopes via step 1 to match additional similar pairs with similar environments. Each residue is allowed to appear in only one residue pair.

Step 3: Continue comparing until all the neighboring residues of "seed 1" are screened or the Pearson correlation coefficient drops to a certain level (<0.5).

Step 4: Outside the seeding distance of seed 1, repeat from step 1 to start a new round of seed identification until all the epitope residues are screened or the Pearson correlation coefficient drops below 0.5.

Until this point, two epitopes were aligned by a group of residue pairs with similar local environment and physicochemical properties. For the highly different epitopes with no similar residue pairs of Pearson correlation coefficients above 0.5, the alignments will still be made according to the ranking of the Pearson correlation coefficients.

**Similarity score of the aligned epitopes**. The similarity score is designed to cover three important measurements: the number of the matched residue pairs ($RP_{AB}$), the environmental similarity ($ES_{AB}$), and the evolutionarily distance as denoted by the residue-transition score ($RT_{AB}$). Then, a linear model is adopted to integrate the above values into a Raw Score ($RS_{AB}$), as shown by formula (1):

$$\begin{cases} RP_{AB} = n \\ ES_{AB} = \sum_1^n P_{k_i l_i} \\ RT_{AB} = \sum_1^n BLOSUM_{62}(k_i, l_i) \\ RS_{AB} = \alpha \cdot RP_{AB} + \beta \cdot ES_{AB} + \gamma \cdot RT_{AB} \end{cases} \quad (1)$$

where, $RP_{AB}$ indicates the number of the matched residue pairs between epitope A and B; $ES_{AB}$ is calculated by accumulating the Pearson correlation coefficients $P_{k_i l_i}$ of all the matched residue pairs $(k_i, l_i)$ derived from fingerprint comparison; and $RT_{AB}$ is equal to the summarized value in the BLOSUM62 matrix for the two matched residues. The $\alpha$, $\beta$, $\gamma$ values are designed to adjust the magnitude of the unbalanced score, and set as 1, 10, 1, respectively.

Finally, the raw score of similarity ($RS_{AB}$) is normalized by the self-query score into a range of (0~1] to remove the size bias of different epitopes. For a query epitope of A, the final similarity score ($SS_{AB}$) to a targeted B epitope can be calculated as shown below:

$$SS_{AB} = \frac{RS_{AB}}{RS_{AA}} \quad (2)$$

where, the similarity score $SS_{AB}$ is the normalized score of target B against query A. Thus, the score of $SS_{AB}$ and $SS_{BA}$ may be different.

**Application of CE-BLAST to different pathogens**. Influenza: As the mutated HA sequences of influenza are highly similar, the epitope alignment can be derived from the sequence alignment. Furthermore, the property of $N$-glycosylation sites are added into the shell structures of the available physicochemical properties. The numbers of potential $N$-glycosylation sites are counted for each layer of shell. The $N$-glycosylation sites are defined by sequons of Asn-X-Ser/Thr, where X represents any amino acid apart from proline[17]. For inter-pathogen cases such as influenza HA protein, the cutoff of $SS_{AB}$ is defined as 0.9, according to the optimal point in the ROC curve. Then, the theoretical antigenicity distance ($TD_{ab}$) is translated and normalized as formula 3, where $\vartheta^0 = 1$, $\vartheta^{(1-cutoff)} = 4$.

$$TD_{ab=}\vartheta^{(1-SS_{AB})} \quad (3)$$

Dengue virus E protein: To generate the complete data for antigenic mapping, 19 tested strains with undone (empty value) in each line were removed from Table S3, described by Leah et al.[6]. For those with antisera values labeled as <10, we arbitrarily set a value of 5 to simplify the calculation. A total of 28 tested strains remained with antisera values for each. Then, for the each line of the tested strain, titer values were normalized within 0 to 1, by setting the highest antisera value as 1 (Supplementary Data 1). Finally, antigenic mapping was performed by ordinal MDS according to the normalized titers.

Potential cross-reactive epitope scanning between DENV and ZIKV: The E protein epitope scanning between DENV and ZIKV contains three steps. First, surface areas of the E protein were identified. Here, the trimer structure of the dengue virus (PDB: 3j27) was selected as a template. The accessible residues were defined as those amino acids on chain C with area solvent accessible surface (ASA) value over 1 Å$^2$, the ASA values were calculated using Naccess[30]. Then, the surface residues were artificially selected among those accessible residues, as shown in Supplementary Fig. 9.

Secondly, all the surface patches were identified. For each residue R as center, all the surface residues within its neighborhood with certain threshold were defined as its surface patch. After scanning all the residues of E protein, different surface patches could be derived.

Finally, the CE-BLAST score of each corresponding surface patch from DENV and ZIKV was calculated. The two surface patches with CE-BLAST scores above the threshold were defined as the potential CRPs. A total of 20 E protein structures including four for ZIKV and 16 for DENV (4 for each serotype) were used in this study; the corresponding strains with GenBank ID are given in Supplementary Table 7.

Experimental validation of influenza H3 can be found Supplementary Note 5.

**Code availability**. Main algorithms were integrated into CE-BLAST web server and can be accessed at http://badd.tongji.edu.cn/ce_blast/ or http://bidd2.nus.edu.sg/czw/ce_blast/. Other code related with this manuscript is available from authors on request.

**Data availability**. In this paper, the HI data and the corresponding strain names used in section High and robust performance on historical HI data of influenza H3 were collected from reports of international organizations and publications as Supplementary Note 2 described. Artificially consolidated dataset can be found in Supplementary Data 2, and sequences were summarized in Supplementary Data 3. The sequence data used in section Reliable prediction for a new influenza H3 vaccine were listed in Supplementary Table 1, sequence comparison of Con H3 and reference strains can be found in Supplementary Fig. 2. Serum data of DENV viruses used in section Correct prediction of serological topology for DENV subtypes were collected from Table S3 of Katzelnick's work[6], and normalized serum data used in this section can be found in Supplementary Data 1. Strain ID of DENV and ZIKV used in section Successful identification of cross-reactive epitopes between DENV and ZIKV were listed in Supplementary Table 7.

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

## Acknowledgements

This work was supported in part by grants from Ministry of Science and Technology China (2010CB833601), the National Natural Science Foundation of China (31171272), the National Postdoctoral Program for Innovative Talents (BX201600033) and the China Postdoctoral Science Foundation Funded Project (2017M611451). Also, we appreciated that Prof. Yuzong Chen and Xian Zeng, from National University of Singapore, helped us to build the mirror server of CE-BLAST in NUS.

## Author contributions

Q.T.Y. developed the algorithm and wrote the majority of the manuscript. Y.Y.Y., T.L.X., and H.X. constructed the CE-BLAST web server and tools. Q.T.Y. and J.X.Q. designed the in silico validation and wrote the partial of the results part. Y.H., X.Y.Z., and J.Q.X. designed the experimental validation and wrote partial of the manuscript. J.X.Q. collected the influenza data and performed the statistical analysis. Q.T.Y., Y.Y.Y., J.X.Q., and D.F.W. constructed figures. D.F.W., Q.C.Z., C.Z., and K.L.T. helped for the model validation. Z.W.C. and J.Q.X. supervised the whole project and modified the manuscript.
