## [Peer Review File · Nature Communications]

Reviewers' comments:

Reviewer #1 (Remarks to the Author):

The authors presented an antigenicity characterization method based on structural information. They claimed this is the first algorithm generalized for all protein antigens.

1. The structural information will be useful in epitope comparison. Probably structural information alone will not be sufficient to define antigenic distances due to the challenges in structural modeling. It is well known the results from structural modeling for single mutation are no more than 70% (e.g. Modeller). Without the details, it will be difficult to believe such structural information alone will be useful for influenza antigenicity measurement, for which a single mutation can lead to antigenic drift. In addition, within a single antigenic drift event, it is not uncommon that the mutations occur in one or more epitope. The current methods seem to only target a single epitope, and this does not seem to be realistic in influenza study.
2. It will be useful and critical for the authors to illustrate the changes of epitope "fingerprints" in the past four decades. Can these results match those antigenic drift events, which have been well characterized?
3. The authors claimed they collected the HI data from various sources for H3N2 data from 1969 to 2013. There is a long list of sources in the Supplementary Note. It is not clear how many HI tables are actually included? How these data are integrated into their analyses? As we know, the HI data from those listed sources were performed at different time, different lab, and different protocols, especially with different red blood cells. Thus, it is difficult to believe validation results using these datasets will be meaningful. The datasets shall be provided to be available to the users.
4. Even the authors claimed that ConH3 is broadly protective, the experimental results suggest that ConH3 only had high cross activities with one testing strain but not others. Because the authors did not enclose the sequences of ConH3, I can not know the details but it seems that ConH3 sequence could be designed from that particular strain. In addition, it is not common to use mice to characterize influenza antigenicity.
5. There are many publications related to epitope comparison using 3D structures. I would suggest the authors review those literatures and discuss the limitations of current studies and how this can be improved.
6. I would suggest the authors focus on influenza viruses. Many statements in this manuscript were either overstated or confusing. For example, the first statement in the summary statement was confusing.

Reviewer #2 (Remarks to the Author):

In the manuscript, Qiu et al. developed a novel computational method, CE-BLAST, to evaluate the antigenic similarity of mutated antigens based on local structure and physicochemical features of conformational epitopes. Their method presented good performances in two independent testing datasets. What impressed me is that they designed a relatively broad-spectrum vaccine strain for influenza A/H3N2 with this method, and found the predictions agree well with the experimental results. However, more details and evidences are needed to support their methods and conclusions.

1. The authors claimed that CE-BLAST could compare the antigenic similarity of any protein antigens and generalize for all protein antigens. But, It looks like the method is more suitable for mutated antigens with one dominant conformation epitope, while not for general protein antigens

pair. One protein antigen may have more than one epitopes. If the method is used to evaluate the antigenic similarity of two proteins, all potential epitope pairs need to be compared and summarized. Thus, the authors could point out the usage conditions and boundaries of their method clearly.

2. For the scarcity of immune cross-reactivity data, the sequence similarity of the variable (V) regions of antibodies were used a measurement of antigenicity similarity. However, the authors did not present more evidence for the reasonability of using V-regions. Could the authors show if there is a high correlation between the sequence similarity of the variable (V) regions of antibodies and the antigenic variation of antigens, such as, influenza HA proteins.

3. There are several parameters in the CE-BLAST algorithm, such as, the α , β and γ in the formula (1). I'm curious how these parameters are determined if there is no training dataset?

4. According to the description, two classes of features were used in the algorithm, including local structural and physicochemical fingerprints. The performance of the final model was tested and shown in the manuscript, while the contributions of each feature was not shown. To be more solid, it could be better to test the contribution of each feature and to show its necessity in the method.

5. The key idea of the CE-BLAST algorithm is to predict the antigenic similarity or variation based on the conformational structure and physicochemical fingerprints. As far as I know, there is method EADpred, published in a paper named "Correlation of Influenza Virus Excess Mortality with Antigenic Variation: Application to Rapid Estimation of Influenza Mortality Burden", had the similar idea. It's better to discuss the similarity and difference between your algorithm and that method.

6. Please provide more details for your designed HA sequence "Con H3". Such as, the sequence of the antigenic sites in "Con H3", and the comparisons between the "Con H3" and the seven reference strains. Also, you'd better to analyze or discuss the molecular mechanism of cross-reactivity of "Con H3" against FJ02, WI05 and BR07 strains.

Reviewer #3 (Remarks to the Author):

This paper presents a method called CE-BLAST for comparing the antigenic similarity between protein antigens, The method is both interesting and novel (it involves spin-image and shell structure models). The paper is generally well written.

A key issue is the evaluation of CE-BLAST's performance. One evaluation was conducted using "sequence similarity of the variable (V) regions of the corresponding antibodies as a measurement of antigenicity similarity between epitopes". In other words, there are known epitopes, but no assays. It is not self-evident that sequence similarity of variable regions is well correlated with the antigenic similarity of their respective epitopes. This really needs to be established - indeed, there must be many contrary examples (e.g. single mutations leading to antigenic escape; non-contact residues in the variable region that have a negligible impact on antigenicity when substituted; different antibodies binding to more-or-less the same epitope using different binding modes). These potential complications need to be addressed as well as their implications for the robustness of the validation.

A second evaluation of CE-BLAST was undertaken using HI assay data for influenza A/H3N2. In this case there are assays, but no epitopes: we are told that "CE-BLAST achieved a high AUC value, over 0.917, based on the 16 antigenic sites that overlap between Liao's work and Smith's work". These sites are known to be predictive of escape to a good degree of accuracy (at least on legacy data). In this context, one possible objection is that, if CE-BLAST is no more accurate than what can be achieved by taking into account a small set of critical residues, what is the added value of the method? This should be addressed.

The claim that "These algorithms [for predicting antigenicity in influenza H3N2] have greatly reduced the experimental workload and facilitated vaccine development" needs support - if the authors know of specific examples, they should be cited. I am not aware that the work cited is actually used by experimentalists - certainly not by leading groups - and I would be interested to see evidence to the contrary.

The paper says: "For antigens without epitope information, it is suggested to model the structure and derive the epitope residues experimentally or computationally." Conformational B-cell epitope prediction (in the absence of an antibody) is extremely challenging and is arguably not be a well-posed problem (see, for example, Ponomarenko & Bourne, BMC Structural Biology 7.1, 2007). In independent evaluations, methods have almost invariably performed poorly. There appears to be a mismatch between such approaches and what is proposed here in terms of the detection of subtle antigenic variations. The authors need to clarify what is, and is not, achievable when there is no structurally defined epitope to work with.

The paper says: "CE-BLAST was first validated on 309 non-redundant known conformational epitopes". Here the term "non-redundant" needs to be clarified: does it mean non-identical in terms of participating residue types, non-identical in terms of epitope:paratope contacts, or something else?

Reviewers' comments:

Reviewer #1 (Remarks to the Author):

The authors presented an antigenicity characterization method based on structural information. They claimed this is the first algorithm generalized for all protein antigens.

1. The structural information will be useful in epitope comparison. Probably structural information alone will not be sufficient to define antigenic distances due to the challenges in structural modeling. It is well known the results from structural modeling for single mutation are no more than 70% (e.g. Modeller). Without the details, it will be difficult to believe such structural information alone will be useful for influenza antigenicity measurement, for which a single mutation can lead to antigenic drift. In addition, within a single antigenic drift event, it is not uncommon that the mutations occur in one or more epitope. The current methods seem to only target a single epitope, and this does not seem to be realistic in influenza study.

Response:

Thank you for pointing this out.

1) Yes, we agree with you that structural information alone may not be sufficient enough. In CE-BLAST, a series of physic-chemical fingerprints beside structure information were added as a complement to evaluate the antigenic distance. Incorporating additional information will be explored in future version of CE-BLAST. Meanwhile, we tested the potential influence on antigenic distance for modeled structures comparing to crystalized structures, taking HA as an example. It can be found that the classification results show full agreement between modeled structures and experimental ones, indicating the feasibility of CE-BLAST on modeled protein structures. The detailed results are illustrated as below:

a) In total, 24 HA crystal structures (Group C) in PDB were collected for the available 1725 HA sequences in our dataset. Their corresponding modeled structures (Group M) were obtained following the same protocols in our study.

b) Pair-wised antigenic distance was calculated via CE-BLAST using default parameters for structures within group C, within group M (intra-group distance), and between Group G and Group M (inter-group distance) respectively. The results can be found in Figure R1. It can be seen that, the classification results shows 100% agreement between both crystalized-modeled structures and modeled-modeled structures, indicating that using modeled structures shows little influence on CE-BLAST results.

Figure R1. Comparison of antigenic distance for inter and intra groups. In fig.1a, each line represents a modeled HA structures and each column represented a crystal structures. In fig.1b, each line and column represented a modeled HA structures while in fig.1c, each line and column represented a crystal structures. The color on each grid refers to the predicted antigenic distance of the corresponding pairs. Blue indicates the antigenic similar pairs (Predicted antigenic distance < 4), while green and yellow represent the antigenic variants (Predicted antigenic distance > 4).

2) It is true that CE-BLAST is more suitable for mutated antigens with one dominate epitope. However, in the case of HA antigen, all the whole neighboring areas of the 16 sites are actually taken into account, which cover the full length of 330 residues in HA1 under default parameters. In this sense, CE-BLAST can detect the antigenic distance caused by any mutations in whole HA1 protein. Here in this paper, we take 16 sites to calculate fingerprints instead of full length, because of their different contribution to antigenicity variance. Meanwhile, selecting representative sites to calculate fingerprints can significantly reduce the computational load.

2. It will be useful and critical for the authors to illustrate the changes of epitope “fingerprint” in the past four decades. Can these results match those antigenic drift events, which have been well characterized?

This is a great question. As you suggested, we calculated the distinct antigenic clusters for the past 47 years (1968-2014) based 16,672 HA1 sequences of H3N2, and added the below to **Results** part at page 12:

“Predicted antigenic clusters of human influenza A /H3N2 virus

Further, we evaluated whether our fingerprint-based algorithm can predict well-known antigenic clusters during the past four decades. Firstly, we calculated the antigenic distance between all strains in each year to the dominant strain of previous year according to chronological order, then we define a new antigenic cluster if the antigenicity coverage of a new antigenic variant within current year is substantially large, comparing to the dominant strain of its previous year. The dominant strain of first year (1968) was set as A/Hong Kong/1/1968. Then the dominant strain in following year was identified as either the one with the largest antigenicity coverage within the year if no antigenic variants emerge, or the antigenic variant with the

largest antigenicity coverage of that year. Details can be found in method part of *CE-BLAST based antigenic clustering algorithm*.

CE-BLAST was used to construct an antigenic map for 16,672 historical HA proteins to determine the antigenic clusters of influenza A/H3N2 from 1968 to 2014 (Fig 4). Different antigenic clusters are marked into different colours. For illustration, the antigenic distance in Y-axis is only meaningful for neighbouring clusters.

Figure 4. **Antigenic clusters for the past 4 decades (1968-2014).** Y-axis illustrated the antigenic distance predicted by CE-BLAST. Each spot represents the dominate strain in that year, with the size in proportional to the logarithmic to the number of strains collected in that year. Antigenically similar years were grouped into antigenic clusters represented by the first dominant strain in the beginning year of each cluster (cluster representative strain). Within each cluster, the antigenic distance was calculated from the dominate strain of each year to the cluster representative strain, while the antigenic distance between two neighbouring clusters was calculated based on the representative strains.

It can be seen that during the past 47 years, 14 antigenic drift events were suggested by our fingerprints. This result agrees well with the well-known experimental study of Smith¹ and *in silico* prediction of Du².

From year 1968 to 2003, 11 antigenic groups were experimentally determined for 273 viral isolates in Smith's work¹, with each named after the first vaccine strain. In our results, these 11 representative vaccine strains were all classified into distinctive clusters. In addition, an additional antigenic group, represented by HK82 (1983-

1986), was detected by our method. Although the experimental screening didn't detect this cluster, Du's study of large-scale antigenicity mapping² pointed out its existence. Du's model predicted the antigenic clusters for H3N2 based on 1071 HA sequences of H3N2 from year 1968 to 2010, mostly from China CDC, and suggested 15 clusters. Compared to our results, 13 clusters are roughly consistent according to time period, while two clusters of CA04 and JX06 were grouped into one of CA04 by CE-BLAST. The high concordance between the two large scale in-silico works may imply the typical epitome of China in global H3N2 profiles. ”

Also, we added a new paragraph “**CE-BLAST based antigenic clustering algorithm**” in the *ONLINE METHODS* part as follows in page 21.

“CE-BLAST based antigenic clustering algorithm

For a year with N strains collected, the antigenic distance was firstly calculated between all the binary strain pairs. Then the antigenic coverage of strain X was defined as $C_X = \frac{N_X}{N}$, where N_X represents the number of antigenically similar pairs (antigenic distance <4) of strain X within the year.

For the first year of 1968 (Y_0), A/Hong Kong/1/1968 was set as the dominant strain (X_0). For next year Y_1 , if a strain X_1 drifts from strain X_0 (antigenic distance >4) and simultaneously has a sufficient large antigenicity population ($C_X > 30\%$) in year Y_1 , X_1 will be determined as the dominant strain of Y_1 , and also the new representative strain of a new cluster to replace X_0 ; otherwise, the dominate strain of year Y_1 is defined as strain X with the highest antigenic coverage C_X in Y_1 , and Y_1 remains in the same cluster as the previous year.”

3. The authors claimed they collected the HI data from various sources for H3N2 data from 1969 to 2013. There is a long list of sources in the Supplementary Note. It is not clear how many HI tables are actually included? How these data are integrated into their analyses? As we know, the HI data from those listed sources were performed at different time, different lab, and different protocols, especially with different red blood cells. Thus, it is difficult to believe validation results using these datasets will be meaningful. The datasets shall be provided to be available to the users.

We agree that HI data from different sources may show different results. Despite that, HI assay is widely used to assess the antigenic similarity between influenza viruses. Maybe we didn't make it clear. Normalization of the experimental distance was done as follows to reduce the data noises:

1. For two virus strains (a and b), we only collect those pairs with all 4 values (H_{aa} , H_{ab} , H_{ba} , H_{bb}) available in one HI table from the same platform. By comparing with self-titration, the result of experimental distance (D_{ab}) is normalized by formula (1).

$$D_{ab} = \log \sqrt{\frac{H_{aa}H_{bb}}{H_{ab}H_{ba}}} \quad (1)$$

2. Different D_{ab} collected from different HI tables are further normalized for each strain pair. For the different D_{ab} values derived from different reports of same strain pair, the outlier D_{ab} will be removed if $|D_{ab} - \overline{D_{ab}}|$ is ranked into top 10% in descending order. The final experimental D_{ab} for each pair is defined as the average of those remained D_{ab} s. Two viruses are defined as antigenic variants when the $\log^{-1}D_{ab}$ is above 4. Otherwise, the pair is treated as antigenic similar.

We checked all the 3867 strain pairs in our dataset. 24 pairs (<1%) were found with contradicting antigenic classification in different experiments (Table 1). The final classification was determined after Step 2, as being shown in Table R1.

In this revised version, HI tables used in this study, and revised method to normalize HI table have been provided in supplementary note 3.

Table R1. Strain pairs with different antigenic relationship from different assays. The first two columns represents two strain pairs with HI assays, then the following three columns represents the number of experimental records with different antigenic relationship.

Strain1	Strain2	>4	=4	<4	Final
A/BANGKOK/1/1979	A/BANGKOK/2/1979	2	0	1	Drift
A/BANGKOK/1/1979	A/MISSISSIPPI/1/1985	2	0	1	Drift
A/BANGKOK/1/1979	A/TEXAS/1/1977	2	2	1	Drift
A/BEIJING/32/1992	A/BEIJING/353/1989	2	0	1	Drift
A/BEIJING/353/1989	A/HONG_KONG/34/1990	2	0	1	Drift
A/CHRISTCHURCH/28/2003	A/FINLAND/486/2004	2	2	1	Drift
A/CHRISTCHURCH/28/2003	A/SHANTOU/1219/2004	2	2	1	Similar
A/ENGLAND/394/2008	A/WISCONSIN/3/2007	1	0	1	Drift
A/ENGLAND/42/1972	A/HONG_KONG/1/1968	6	0	5	Drift
A/ENGLAND/878/1969	A/HONG_KONG/1/1968	3	1	1	Drift
A/FINLAND/9/2008	A/WISCONSIN/67/2005	1	1	6	Similar
A/FUJIAN/140/2000	A/SYDNEY/05/1997	3	2	1	Drift
A/HIROSHIMA/33/2006	A/WISCONSIN/67/2005	1	0	1	Similar
A/LYON/636/2006	A/WISCONSIN/67/2005	1	1	4	Similar
A/MISSISSIPPI/1/1985	A/PHILIPPINES/2/1982	1	0	2	Drift
A/MOSCOW/10/1999	A/PANAMA/2007/1999	1	0	14	Similar
A/MOSCOW/10/1999	A/SYDNEY/05/1997	1	0	10	Similar
A/PANAMA/2007/1999	A/SYDNEY/05/1997	1	0	10	Similar
A/VICTORIA/523/2004	A/WISCONSIN/67/2005	1	0	1	Drift
A/PERTH/16/2009	A/VICTORIA/361/2011	18	22	21	Drift
A/ALABAMA/05/2010	A/VICTORIA/361/2011	5	2	50	Similar
A/PERTH/10/2010	A/VICTORIA/361/2011	2	0	2	Drift
A/PERTH/16/2009	A/TEXAS/50/2012	13	20	1	Drift
A/ALABAMA/05/2010	A/HAWAII/22/2012	7	14	10	Similar

4. Even the authors claimed that ConH3 is broadly protective, the experimental results suggest that ConH3 only had high cross activities with one testing strain but not

others. Because the authors did not enclose the sequences of ConH3, I can not know the details but it seems that ConH3 sequence could be designed from that particular strain. In addition, it is not common to use mice to characterize influenza antigenicity.

In our manuscript, the main purpose of Con H3 case was designed to illustrate the usefulness of CE-Blast in facilitating future vaccine design. The validated experimental results fully agreed with our predicted results. Although relatively lower in titers when compared to WI05 H3 group, the titers against BR07 H3 and FJ02 H3 strains are fairly strong when compared with that in control groups. In addition, the antibody binding assay in Figure 3 shows the cross-activity to BR07 H3 and FJ02 H3 strains.

The sequence of “Con H3” is designed from collected strains from 2006 to 2009. The sequence of “Con H3” is given as follows:

MKTIIALSYILCLVFAQKLPNGNDNSTATLCLGHHAVPNGTIVKTITNDQIEVTNA
TELVQSSSTGEICDSPHQILDGENCTLIDALLGDPQCDGFQNKKWDLFVERSK
AYSNCYPYDVPDYASLRSLVASSGTLEFNNEFNWTGVTQNGTSSACIRRSNN
SFFSRLNWLTHLKFYPALNVTMPNNEQFDKLYIWGVHHPGTDNDQIFLYAQ
ASGRITVSTKRSQQTVIPNIGSRPRVRNIPSRISYWTIVKPGDILLINSTGNLIAP
RGYFKIRSGKSSIMRSDAPIGKCNSECITPNGSIPNDKPFQNVNRITYGACPRYV
KQNTLKLATGMRNVPEKQTRGIFGAIAGFIENGWEGMVDGWYGRHQNSEG
RGQAADLKSTQAAIDQINGKLNRLIGKTNEKFHQIEKEFSEVEGRIQDLEKYV
EDTKIDLWSYNAELLVALENQHTIDLTSEMKNLFEKTKKQLRENAEDMGNG
CFKIYHKCDNACIGSIRNGTYDHDVYRDEALNNRFQIKGVELKSGYKDWILW
ISFAISCFLLCVALLGFIMWACQKGNIRCNICI

Although ferret is more frequently used to assess influenza antigenicity, a number of reports have recently adopted mice¹⁻⁵ to evaluate vaccine immunogenicity and efficacy as mice model is much more cost-effective and readily available; In addition, the mice is capable to mount vigorous both humoral and T-cell immune responses.

1. Sui J, et al. Protective efficacy of an inactivated Eurasian avian-like H1N1 swine influenza vaccine against homologous H1N1 and heterologous H1N1 and H1N2 viruses in mice. *Vaccine*. 2016 Jul 19;34(33):3757-63.
2. Kamlangdee A, et al. Mosaic H5 Hemagglutinin Provides Broad Humoral and Cellular Immune Responses against Influenza Viruses. *J Virol*. 2016 Jul 11;90(15):6771-83.
3. Hoa le NM, et al. Association between Hemagglutinin Stem-Reactive Antibodies and Influenza A/H1N1 Virus Infection during the 2009 Pandemic. *J Virol*. 2016 Jun 24;90(14):6549-56.
4. Venereo-Sanchez A, et al. Hemagglutinin and neuraminidase containing virus-like particles produced in HEK-293 suspension culture: An effective influenza vaccine candidate. *Vaccine*. 2016 Jun 17;34(29):3371-80.

5. Tan GS, et al. Broadly-Reactive Neutralizing and Non-neutralizing Antibodies Directed against the H7 Influenza Virus Hemagglutinin Reveal Divergent Mechanisms of Protection. *PLoS Pathog.* 2016 Apr 15;12(4):e1005578.

5. There are many publications related to epitope comparison using 3D structures. I would suggest the authors review those literatures and discuss the limitations of current studies and how this can be improved.

Thank you for pointing this out. Currently, there are several methods published for epitope comparison using 3D structures. We compared three available methods: EpitopeMatch, EADpred and PREDAC, and added below paragraph into *Discussion* part of our revised manuscript at page 14:

“In addition to structure comparison, there are a few methods related to epitope comparison using 3D structures such as EpitopeMatch, EADpred and PREDAC. In 2009, EpitopeMatch was firstly developed to superimpose structural epitopes to derive RMSD based on atom coordinates and atom types without consideration of physic-chemical properties of amino acids. Then Jiang’s group designed a machine learning method of EADpred for H1N1 via comparing the averaged difference for 6 epitope areas based on individual physic-chemical property of HA residues. The idea was further expanded into PREDAC for H3N2 antigenic clustering, incorporating structural distance to “receptor binding” regions. In this paper, CE-BLAST takes full account of the potential influence from 3-D residual layout and local micro-environment caused by structural variation and residual mutation.”

6. I would suggest the authors focus on influenza viruses. Many statements in this manuscript were either overstated or confusing. For example, the first statement in the summary statement was confusing.

The confusing statement has been corrected in this version. We revised the summary paragraph in the *Discussion* part as follows at page 15:

“In conclusion, we designed a generalized algorithm that is able to compare the antigenicity similarity between different conformational epitopes of protein antigens. The usefulness of CE-BLAST has been shown on influenza viruses because of the vast accumulation of HA mutant data. Other types of mutated antigens are also feasible once the epitope positions defined. Subsequent improvements can be further elaborated on additional parameters, and parallel computing to increase the computational efficiency. With future updating of our epitope database to cover HIV and tumor antigens, CE-BLAST may become more useful in facilitating the antigenicity mapping and potential vaccine development for a wide range of diseases.”

Reviewer #2 (Remarks to the Author):

In the manuscript, Qiu et al. developed a novel computational method, CE-BLAST, to evaluate the antigenic similarity of mutated antigens based on local structure and physicochemical features of conformational epitopes. Their method presented good performances in two independent testing datasets. What impressed me is that they designed a relatively broad-spectrum vaccine strain for influenza A/H3N2 with this method, and found the predictions agree well with the experimental results. However, more details and evidences are needed to support their methods and conclusions.

1. The authors claimed that CE-BLAST could compare the antigenic similarity of any protein antigens and generalize for all protein antigens. But, It looks like the method is more suitable for mutated antigens with one dominant conformation epitope, while not for general protein antigens pair. One protein antigen may have more than one epitopes. If the method is used to evaluate the antigenic similarity of two proteins, all potential epitope pairs need to be compared and summarized. Thus, the authors could point out the usage conditions and boundaries of their method clearly.

Yes, you are right.

“Theoretically, CE-BLAST is designed to compare any two pieces of conformational epitopes, no matter they are from the same protein family or not. In fact, it is more suitable for mutated antigens as the similarity score can be high enough to indicate the antigenic similarity. If a protein antigen has more than one epitopes, all potential epitope pairs need to be compared and summarized in order to evaluate the antigenic similarity of two proteins.”

We have added above statement into *Discussion* part at Page 15.

2. For the scarcity of immune cross-reactivity data, the sequence similarity of the variable (V) regions of antibodies were used a measurement of antigenicity similarity. However, the authors did not present more evidence for the reasonability of using V-regions. Could the authors show if there is a high correlation between the sequence similarity of the variable (V) regions of antibodies and the antigenic variation of antigens, such as, influenza HA proteins.

This is a smart question. For other antigens without immune cross-reactivity data, the similarity of corresponding antibody’s V-region need to be checked as a measurement of antigenicity similarity. We have added the below part into *Results* at page 6:

“To check the qualification as a measurement of antigenicity similarity, the sequence similarity of the variable (V) regions of antibodies were examined in correlation with the variance of epitope compositions. Lysozyme was chosen as an example under test since it is the largest protein family in our dataset.

There are total 40 lysozyme-antibody complexes in PDB. The size of lysozyme

epitopes is on average of 26 ± 4 residues. The Pearson correlation coefficient between epitope compositional difference and V-region similarity of corresponding antibodies can reach 0.7346 (identities). From the literature recording, antibodies are derived from different immune host such as mus (28 complexes), camelus (6 complexes), homo sapiens (2 complexes) and others. Considering the V-sequence bias of different immune host, we take the 28 complexes from mice for further examine. The correlation between was then increased to 0.8725 (antibody positive similarity) and 0.8666 (antibody identity similarity), as being shown in Supplementary Figure 1. The high correlation above suggests the feasibility of using V-region sequence similarity of antibodies as a measurement of antigenicity similarity. However, we also noted that the same conformational epitope of lysozyme may induce quite different sequences of V-region antibodies from different immune host. For instance, the sequence identify score of V-region is only 75% between human antibody (2EIZ) and mice antibody (1XGP) to the identical lysozyme epitope. This seems to indicate that it is more meaningful to adopt V-region sequence similarity of antibodies as a measurement of antigenicity similarity in the case of same immune host.”

Supplementary Figure 1. Correlations between number of different residues and V-region similarity of corresponding antibodies. Red scatter diagram represent V-region similarity (positive). Blue scatter diagram represent V-region similarity (identities).

3. There are several parameters in the CE-BLAST algorithm, such as, the α , β and γ in the formula (1). I'm curious how these parameters are determined if there is no training dataset?

The raw score was determined by $RS_{AB} = \alpha \cdot RP_{AB} + \beta \cdot ES_{AB} + \gamma \cdot RT_{AB}$ in formula (1). Here, RP_{AB} indicates residual pairs, and RT_{AB} indicates residual

translational score. They are both real numbers. In contrast, ES_{AB} represents the accumulation of Pearson correlation coefficient which ranges from -1 to 1. To make the three values at the same level of magnitude order, β was arbitrarily set as 10 while α and γ were set as 1.

4. According to the description, two classes of features were used in the algorithm, including local structural and physicochemical fingerprints. The performance of the final model was tested and shown in the manuscript, while the contributions of each feature was not shown. To be more solid, it could be better to test the contribution of each feature and to show its necessity in the method.

This is a good question. We tested the contribution of each features on different protein families, such as 40 lysozyme complexes and 23 GP120 complexes. It can be seen in table 2 that the contribution of each feature actually varies according to different protein families. Nevertheless, combined features always give the best performance. Thus we used the combined features in our model.

Table 2. Contributions of each feature

	Lysozyme(I98) ^a	Lysozyme(I100) ^b	GP120(I98)	GP120(I100)
Combined^c	0.974	0.933	0.950	0.950
Structure	0.958	0.912	0.781	0.776
Hydrophobic	0.934	0.817	0.945	0.944
H-bond only	0.959	0.792	0.947	0.948
Electrostatic only	0.967	0.887	0.946	0.947
Generalized Physical-chemical^d	0.952	0.860	0.948	0.949

^aI98 means the V-region similarity is set as 98% identities.

^bI100 means the V-region similarity is set as 100% identities.

^cCombined means structure fingerprints combined with physical-chemical fingerprints.

^dGeneralized Physical-chemical means combined above three physical-chemical properties.

5. The key idea of the CE-BLAST algorithm is to predict the antigenic similarity or variation based on the conformational structure and physicochemical fingerprints. As far as I know, there is method EADpred, published in a paper named "Correlation of Influenza Virus Excess Mortality with Antigenic Variation: Application to Rapid Estimation of Influenza Mortality Burden", had the similar idea. It's better to discuss the similarity and difference between your algorithm and that method.

Thank you for reminding. We added one paragraph in the *Discussion* part of our revised manuscript as follows at page 14:

“In addition to structure comparison, there are a few methods related to epitope comparison using 3D structures such as EpitopeMatch, EADpred and PREDAC. In 2009, EpitopeMatch was firstly developed to superimpose structural epitopes to derive RMSD based on atom coordinates and atom types without consideration of physic-chemical properties of amino acids. Then Jiang’s group designed a machine learning method of EADpred for H1N1 via comparing the averaged difference for 6

epitope areas based on individual physic-chemical property of HA residues. The idea was further expanded into PREADAC for H3N2 antigenic clustering, incorporating structural distance to “receptor binding” regions. In this paper, CE-BLAST takes full account of the potential influence from 3-D residual layout and local micro-environment caused by structural variation and residual mutation.”

6. Please provide more details for your designed HA sequence "Con H3". Such as, the sequence of the antigenic sites in "Con H3", and the comparisons between the "Con H3" and the seven reference strains. Also, you'd better to analyze or discuss the molecular mechanism of cross-reactivity of "Con H3" against FJ02, WI05 and BR07 strains.

As suggested, we provided "Con H3" sequence and compared it to the seven reference strains. We further added the following paragraph into the section of Results on page 11. As it says:

“Further, we compared the HA1 sequence between “Con H3” and seven reference strains. The results are showed in Supplementary figure 6. Compared with “Con H3”, A/Brisbane/10/2007 was identified two mutations in HA1 sequence with one positioned within 16 antigenic sites, and A/Wisconsin/67/2005 contains seven mutations also with one in 16 antigenic sites. Similarly A/Fujian/411/2002 possesses 7 mutations with two in 16 antigenic sites.

Interestingly, at least 24 mutations were identified for the other 4 antigenically varied strains. This may explain the differences in cross-reactivities among them.”

```

CON_H3
A/Brisbane/10/2007
A/Wisconsin/67/2005
A/Fujian/411/2002
A/Hong_Kong/1/1968
A/Philippines/2/1982
A/Moscow/10/1999
A/Indiana/08/2011

1         10        20        30        40        50        60
QKLPGNDNSTATLCLGHHAVPNGTIVKTIITNDQIEVTNATELVQSSSTGECDSPHQILD
QKLPGNDNSTATLCLGHHAVPNGTIVKTIITNDQIEVTNATELVQSSSTGECDSPHQILD
QKLPGNDNSTATLCLGHHAVPNGTIVKTIITNDQIEVTNATELVQSSSTGECDSPHQILD
QKLPGNDNSTATLCLGHHAVPNGTIVKTIITNDQIEVTNATELVQSSSTGECDSPHQILD
QKLPGNDNSTATLCLGHHAVPNGTIVKTIITNDQIEVTNATELVQSSSTGECDSPHQILD
QKLPGNDNSTATLCLGHHAVPNGTIVKTIITNDQIEVTNATELVQSSSTGECDSPHQILD
QKLPGNDNSTATLCLGHHAVPNGTIVKTIITNDQIEVTNATELVQSSSTGECDSPHQILD
QKLPGNDNSTATLCLGHHAVPNGTIVKTIITNDQIEVTNATELVQSSSTGECDSPHQILD
QKLPGNDNSTATLCLGHHAVPNGTIVKTIITNDQIEVTNATELVQSSSTGECDSPHQILD
QKLPGNDNSTATLCLGHHAVPNGTIVKTIITNDQIEVTNATELVQSSSTGECDSPHQILD

61        70        80        90        100       110       120
GENCTLIDALLGDPQCDGFGQKQKNDLDFVERSKAYSNCYFPYDVPDYASLRSLVASSGTLEF
GENCTLIDALLGDPQCDGFGQKQKNDLDFVERSKAYSNCYFPYDVPDYASLRSLVASSGTLEF
GENCTLIDALLGDPQCDGFGQKQKNDLDFVERSKAYSNCYFPYDVPDYASLRSLVASSGTLEF
GENCTLIDALLGDPQCDGFGQKQKNDLDFVERSKAYSNCYFPYDVPDYASLRSLVASSGTLEF
GENCTLIDALLGDPQCDGFGQKQKNDLDFVERSKAYSNCYFPYDVPDYASLRSLVASSGTLEF
GENCTLIDALLGDPQCDGFGQKQKNDLDFVERSKAYSNCYFPYDVPDYASLRSLVASSGTLEF
GENCTLIDALLGDPQCDGFGQKQKNDLDFVERSKAYSNCYFPYDVPDYASLRSLVASSGTLEF
GENCTLIDALLGDPQCDGFGQKQKNDLDFVERSKAYSNCYFPYDVPDYASLRSLVASSGTLEF
GENCTLIDALLGDPQCDGFGQKQKNDLDFVERSKAYSNCYFPYDVPDYASLRSLVASSGTLEF
GENCTLIDALLGDPQCDGFGQKQKNDLDFVERSKAYSNCYFPYDVPDYASLRSLVASSGTLEF

121       130       140       150       160       170       180
NNRFRNWTGVTQNGCTSSACIRRSNNSFFSRLNWLTHLKFNYPALNVTPNNPDKLYIN
NNRFRNWTGVTQNGCTSSACIRRSNNSFFSRLNWLTHLKFNYPALNVTPNNPDKLYIN
NNRFRNWTGVTQNGCTSSACIRRSNNSFFSRLNWLTHLKFNYPALNVTPNNPDKLYIN
NNRFRNWTGVTQNGCTSSACIRRSNNSFFSRLNWLTHLKFNYPALNVTPNNPDKLYIN
NNRFRNWTGVTQNGCTSSACIRRSNNSFFSRLNWLTHLKFNYPALNVTPNNPDKLYIN
NNRFRNWTGVTQNGCTSSACIRRSNNSFFSRLNWLTHLKFNYPALNVTPNNPDKLYIN
NNRFRNWTGVTQNGCTSSACIRRSNNSFFSRLNWLTHLKFNYPALNVTPNNPDKLYIN
NNRFRNWTGVTQNGCTSSACIRRSNNSFFSRLNWLTHLKFNYPALNVTPNNPDKLYIN
NNRFRNWTGVTQNGCTSSACIRRSNNSFFSRLNWLTHLKFNYPALNVTPNNPDKLYIN
NNRFRNWTGVTQNGCTSSACIRRSNNSFFSRLNWLTHLKFNYPALNVTPNNPDKLYIN

181       190       200       210       220       230       240
GVVHPGTNDQIQIFLYAQASGRITVSTKRSQQTVIPNIGSRPRVRNIPSRISYWTIVKPG
GVVHPGTNDQIQIFLYAQASGRITVSTKRSQQTVIPNIGSRPRVRNIPSRISYWTIVKPG
GVVHPGTNDQIQIFLYAQASGRITVSTKRSQQTVIPNIGSRPRVRNIPSRISYWTIVKPG
GVVHPGTNDQIQIFLYAQASGRITVSTKRSQQTVIPNIGSRPRVRNIPSRISYWTIVKPG
GVVHPGTNDQIQIFLYAQASGRITVSTKRSQQTVIPNIGSRPRVRNIPSRISYWTIVKPG
GVVHPGTNDQIQIFLYAQASGRITVSTKRSQQTVIPNIGSRPRVRNIPSRISYWTIVKPG
GVVHPGTNDQIQIFLYAQASGRITVSTKRSQQTVIPNIGSRPRVRNIPSRISYWTIVKPG
GVVHPGTNDQIQIFLYAQASGRITVSTKRSQQTVIPNIGSRPRVRNIPSRISYWTIVKPG
GVVHPGTNDQIQIFLYAQASGRITVSTKRSQQTVIPNIGSRPRVRNIPSRISYWTIVKPG
GVVHPGTNDQIQIFLYAQASGRITVSTKRSQQTVIPNIGSRPRVRNIPSRISYWTIVKPG

241       250       260       270       280       290       300
DILLINSTGNLIAPRGYFKIRSGKSSIMRSDAPICCNSECITPNGSIPNDKPFQNVNRI
DILLINSTGNLIAPRGYFKIRSGKSSIMRSDAPICCNSECITPNGSIPNDKPFQNVNRI
DILLINSTGNLIAPRGYFKIRSGKSSIMRSDAPICCNSECITPNGSIPNDKPFQNVNRI
DILLINSTGNLIAPRGYFKIRSGKSSIMRSDAPICCNSECITPNGSIPNDKPFQNVNRI
DILLINSTGNLIAPRGYFKIRSGKSSIMRSDAPICCNSECITPNGSIPNDKPFQNVNRI
DILLINSTGNLIAPRGYFKIRSGKSSIMRSDAPICCNSECITPNGSIPNDKPFQNVNRI
DILLINSTGNLIAPRGYFKIRSGKSSIMRSDAPICCNSECITPNGSIPNDKPFQNVNRI
DILLINSTGNLIAPRGYFKIRSGKSSIMRSDAPICCNSECITPNGSIPNDKPFQNVNRI
DILLINSTGNLIAPRGYFKIRSGKSSIMRSDAPICCNSECITPNGSIPNDKPFQNVNRI
DILLINSTGNLIAPRGYFKIRSGKSSIMRSDAPICCNSECITPNGSIPNDKPFQNVNRI

301       310       320       329
TYGACFRYVKQNTLELATGMRNVPEKQTR
TYGACFRYVKQNTLELATGMRNVPEKQTR
TYGACFRYVKQNTLELATGMRNVPEKQTR
TYGACFRYVKQNTLELATGMRNVPEKQTR
TYGACFRYVKQNTLELATGMRNVPEKQTR
TYGACFRYVKQNTLELATGMRNVPEKQTR
TYGACFRYVKQNTLELATGMRNVPEKQTR
TYGACFRYVKQNTLELATGMRNVPEKQTR
TYGACFRYVKQNTLELATGMRNVPEKQTR
TYGACFRYVKQNTLELATGMRNVPEKQTR

```

Supplementary Figure 6. Sequence comparison of “Con H3” and seven reference strains. Mutations of reference strains compared with “Con H3” were shadowed in different color. The 16 antigenic sites used in CE-BLAST were marked in red box.

Reviewer #3 (Remarks to the Author):

This paper presents a method called CE-BLAST for comparing the antigenic similarity between protein antigens. The method is both interesting and novel (it involves spin-image and shell structure models). The paper is generally well written.

A key issue is the evaluation of CE-BLAST's performance. One evaluation was conducted using "sequence similarity of the variable (V) regions of the corresponding antibodies as a measurement of antigenicity similarity between epitopes". In other

words, there are known epitopes, but no assays. It is not self-evident that sequence similarity of variable regions is well correlated with the antigenic similarity of their respective epitopes. This really needs to be established - indeed, there must be many contrary examples (e.g. single mutations leading to antigenic escape; non-contact residues in the variable region that have a negligible impact on antigenicity when substituted; different antibodies binding to more-or-less the same epitope using different binding modes). These potential complications need to be addressed as well as their implications for the robustness of the validation.

Thanks for raising this question. It is really worth to examine whether the V-region similarity of antibodies may correlate with the antigenicity similarity of respective epitopes, when no assays are available. We have added below to the **Results** part of revised manuscript at page 6:

“To check the qualification as a measurement of antigenicity similarity, the sequence similarity of the variable (V) regions of antibodies were examined in correlation with the variance of epitope compositions. Lysozyme was chosen as an example under test since it is the largest protein family in our dataset.

There are total 40 lysozyme-antibody complexes in PDB. The size of lysozyme epitopes is on average of 26 ± 4 . The Pearson correlation coefficient between epitope compositional difference and V-region similarity of corresponding antibodies can reach 0.7346 (identities). From the literature recording, antibodies are derived from different immune host such as mice (28 complexes), camelus (6 complexes), homo sapiens (2 complexes) and others. Considering the V-sequence bias of different immune host, we take the 28 complexes from mice for further examine. The correlation between epitope compositional difference and the V-region sequence similarity was increased to 0.8725 (antibody positive similarity) and 0.8666 (antibody identity similarity), as being shown in Supplementary Figure 1. The high correlation above suggests the feasibility of using V-region sequence similarity of antibodies as a measurement of antigenicity similarity. However, we also noted that the same conformational epitope of lysozyme may induce quite different V-region antibodies from different immune host. For instance, the sequence identify score of V-region is only 75% between human antibody (2EIZ) and mice antibody (1XGP) to the same lysozyme epitope. This seems to indicate that it is more meaningful to adopt V-region sequence similarity of antibodies as a measurement of antigenicity similarity in the case of same immune host.”

Supplementary Figure 1. Correlations between number of different residues and V-region similarity of corresponding antibodies. Red scatter diagram represent V-region similarity (positive). Blue scatter diagram represent V-region similarity (identities).

A second evaluation of CE-BLAST was undertaken using HI assay data for influenza A/H3N2. In this case there are assays, but no epitopes: we are told that "CE-BLAST achieved a high AUC value, over 0.917, based on the 16 antigenic sites that overlap between Liao's work and Smith's work". These sites are known to be predictive of escape to a good degree of accuracy (at least on legacy data). In this context, one possible objection is that, if CE-BLAST is no more accurate than what can be achieved by taking into account a small set of critical residues, what is the added value of the method? This should be addressed.

Your question is reasonable. First of all, CE-BLAST is designed for all types of protein antigens, not just for influenza only. Validation through HI assay data on influenza is meant to test the generalizability of the application scope.

In the context of influenza, the huge accumulation of historical data has derived a small set of critical residues for various predictive methods to a certain degree of accuracy. However, a number of reports pointed out that, these critical positions are actually evolving with time in HA antigen¹⁻⁵. Antigenic variants caused by new mutations out of the pre-defined critical positions may not be detected by traditional site-dependent methods. While in CE-BLAST, no training data is required. The 16 critical positions are only used as the anchors to scan the whole structure of HA protein. Any mutations in whole HA1 sequence will be calculated for its antigenicity contribution.

At last, we compared the performance of CE-BLAST to the intuitive method by

similarity score on these 16 antigenic sites. The similarity score between strains pair is defined as $SS = 1 - \frac{N_m}{16}$ (N_m indicates the number of mutations on 16 sites). The classification results by SS only reach the AUC value of 0.6422 on our 3867 historical dataset.

1. Bush, R.M, et al. Predicting the evolution of human influenza A. *Science* 286, 1921-1925 (1999).
2. Smith, D.J. et al. Mapping the antigenic and genetic evolution of influenza virus. *Science* 305, 371-376 (2004).
3. Lee, M.S. et al. Predicting antigenic variants of influenza A/H3N2 viruses. *Emerging infectious diseases* 10, 1385-1390 (2004).
4. Liao, Y.C. et al. ATIVS: analytical tool for influenza virus surveillance. *Nucleic acids research* 37, W643-646 (2009).
5. Koel, B.F. et al. Substitutions near the receptor binding site determine major antigenic change during influenza virus evolution. *Science* 342, 976-979 (2013)

The claim that "These algorithms [for predicting antigenicity in influenza H3N2] have greatly reduced the experimental workload and facilitated vaccine development" needs support - if the authors know of specific examples, they should be cited. I am not aware that the work cited is actually used by experimentalists - certainly not by leading groups - and I would be interested to see evidence to the contrary.

Thank you for pointing this out. We have modified our statement in ***Introduction*** Part of our revised manuscript as follows at page 2:

“These algorithms have been explored to assist influenza surveillance and potential vaccine strain recommendation¹⁻³”

1. Pan, K, et al. A novel sequence-based antigenic distance measure for H1N1, with application to vaccine effectiveness and the selection of vaccine strains. *Protein engineering design & selection*. 24 (3): 291-299 (2011).
2. Ndifon, W, et al. Differential neutralization efficiency of hemagglutinin epitopes, antibody interference, and the design of influenza vaccines. *PNAS*. 106 (21): 8701-8706 (2009).
3. Xiangjun Du, et al. Mapping of H3N2 influenza antigenic evolution in China reveals a strategy for vaccine strain recommendation. *Nature Communications*. 3, 709 (2012).

The paper says: "For antigens without epitope information, it is suggested to model the structure and derive the epitope residues experimentally or computationally." Conformational B-cell epitope prediction (in the absence of an antibody) is extremely challenging and is arguably not be a well-posed problem (see, for example, Ponomarenko & Bourne, BMC Structural Biology 7.1, 2007). In independent evaluations, methods have almost invariably performed poorly. There appears to be a mismatch between such approaches and what is proposed here in terms of the detection of subtle antigenic variations. The authors need to clarify what is, and is not, achievable when there is no structurally defined epitope to work with.

We fully agree with your opinion. The sentences have been modified as below in **Results** part at page 5:

“To guarantee the accuracy of CE-BLAST, users are encouraged to use well determined antigenic sites. For antigens without epitope information, available B-cell epitope prediction methods may be explored for consensus sites before testing CE-BLAST.”

The paper says: "CE-BLAST was first validated on 309 non-redundant known conformational epitopes". Here the term "non-redundant" needs to be clarified: does it mean non-identical in terms of participating residue types, non-identical in terms of epitope: paratope contacts, or something else?

Thank you for pointing this out. We have clarified the terms in **METHODS** part as follows at page 17:

“Redundant epitopes were identified only when both the residual composition of epitope and the sequence of corresponding antibody were identical.”

Reviewers' comments:

Reviewer #1 (Remarks to the Author):

Qiu et al. submitted a revised version based on the earlier suggestion. Only some of the comments raised from the last review were addressed and I still have a few major comments for this manuscript.

1. I am still not convinced that this method can be useful in quantifying influenza antigenicity. As the authors agreed, the predicted structure would NOT be accurate enough to measure the antigenic changes from a single mutation in epitope, however, which could lead to an antigenic drift in influenza virus. The results from antigenic analyses of the viruses from the past four decades (Figure 4) do not seem correct. First of all, the antigenic drift events were not marked correctly. For example, this figure missed important antigenic drift events from CA04 to BR07 and that from TX12 to SWZ13. Second, the re-emerging H3N2 viruses were not detected at all. Third, it is not clear how the antigenic distances are associated with those in the serological assays and it is very difficult to believe the useful of this method.

2. The data used in the analyses had too many noises. In the tables listed in supplementary file, the HI data were processed with different protocols. For example, some HI data were processed using turkey red blood cells and some using guinea pig red blood cells. Some HI data were processed with the addition of neuraminidase inhibitors. The authors can not simply use mean titers for their analyses. Overall, current analyses are still too crude to be believed.

3. The blast result to influenza database showed that ConH3 sequence is 100% identical to the HA protein sequence of A/Kentucky/01/2007(H3N2). It would be difficult to image this field strain can generate a broad protective spectrum. The authors may need to understand an influenza virus can cross react with those viruses which are circulated before or after this virus. For example, the H3N2 viruses in 2007 could have cross activities (to different extents) with those viruses in 2004 and 2009. The results showed that the sera from ConH3 had a titer of 12,150 against WI05, 2786 against BR07, 2111 against FJ02, and not for MO99. Thus, this result suggested that the ConH3 is most likely designed from a strain and would be expected to have similar immunity to this strain. In summary, the broad protective spectrum for ConH3 is not convincing.

Reviewer #2 (Remarks to the Author):

I have studied the revised version of the manuscript, with particular emphasis on the reasonable measurement of antigenicity similarity and features' contribution in the model. Given the renewed descriptions and results, I am now convinced of the reasonability of their methodology. I do not have any more comments.

Reviewer #3 (Remarks to the Author):

I think you have done a good job of addressing my original concerns.

Reviewers' comments:

Reviewer #1 (Remarks to the Author):

1. I am still not convinced that this method can be useful in quantifying influenza antigenicity. As the authors agreed, the predicted structure would NOT be accurate enough to measure the antigenic changes from a single mutation in epitope, however, which could lead to an antigenic drift in influenza virus.

We understood your concern. In our analysis, we noticed that the substantial conformational change of protein antigen do have considerable influence on antigenicity prediction, probably because of structural exposure of different epitope areas. CE-BLAST works better for antigens with relatively stable structure and well-defined epitope areas. We have added part of above into *DISCUSSION* section at page 17 as follows:

According to our algorithm, the reasons why CE-BLAST may detect the antigenic change caused by a single mutation in epitope, cover at least 3 aspects: 1) The residual layout difference can be recorded by spin-images of all its neighboring residues around the mutation; 2) The micro-environmental change caused by point mutation can be detected by shells of physical-chemical fingerprints from all its neighboring residues; 3) The substantial penalty will be deducted from similarity score during residual comparison. In brief, the residual layout and physic-chemical change caused by a single mutation can be recorded from different angles by all its neighboring residues. Coupled with the penalty from BLOSUM matrix, the sensitivity to measure the antigenic changes from a single mutation has been largely increased in CE-BLAST.

The results from antigenic analyses of the viruses from the past four decades (Figure 4) do not seem correct. First of all, the antigenic drift events were not marked correctly. For example, this figure missed important antigenic drift events from CA04 to BR07 and that from TX12 to SWZ13. Second, the re-emerging H3N2 viruses were not detected at all.

Thanks for pointing out Figure 4. Figure 4 is generated based on the overall results by all pairs between 1725 representative strains. Each spot of the year actually illustrates the antigenicity distance of major population to either the representative strain of the previous cluster if it is the initiative year of a new cluster, or that of the current cluster if not initiative, instead of individual strain pairs. In figure 4, a new cluster was defined only when the proportion of escaping strains rises up to more than 30% in strain population of current year. Thus, the antigenic drift between strain pairs is not shown in Figure 4.

Regarding your concerns, we feel also bewildered about the drift cases since opposite evidence were found from CDC reports. From CA04 (A/California/7/2004)

to BR07 (A/Brisbane/10/2007), we obtained the experimental results from CDC report (Figure R1 & Table R1). According to formula (1) of $D_{ab} = \log \sqrt{\frac{H_{aa}H_{bb}}{H_{ab}H_{ba}}}$, the experimental antigenic distance (Dab) of CA04 (F859) & BR07 (F968) equals to 2, and Dab of CA04 (F859) & BR07 (F998) equals 2.828. Both of the experimental distances are below 4, indicating the similarity of the two strains, which agree well with our predicted result.

If possible, additional experimental evidence or published data are highly appreciated for us to calibrate if you could kindly provide.

		A	B	C	D	E	F	G
		F859	F854	F897	F948	F966	F968	F998
	Reference Strains	CAL/7	WISCONSIN/7	BRIS/9	NEP/921	BRIS/4	BRIS/10	BRIS/10
A	A/CALIFORNIA/7/2004	1280	320	320	40	640	160	160
B	A/WISCONSIN/7/2005	1280	640	1280	160	1280	640	320
C	A/BRISBANE/9/2006	640	640	1280	160	1280	320	320
D	A/NEPAL/921/2006	320	320	640	160	320	320	320
E	A/BRISBANE/4/2007	640	320	1280	160	1280	640	640
F	A/BRISBANE/10/2007	1280	640	640	160	1280	640	640
G	A/BRISBANE/10/2007	80	<40	80	40	40	160	80
H	A/PERTH/27/2007	160	80	320	80	160	320	320
I	A/PHILIPPINES/1616/2007	80	<40	80	<40	40	80	160
J	A/WISCONSIN/3/2007	1280	640	2560	320	1280	1280	640
K	A/BRISBANE/202/2007	2560	1280	1280	640	2560	2560	2560
L	A/BRISBANE/229/2007	160	320	640	80	640	160	80

Figure-Response 1*. Screenshots from CDC report

Table-Response 1*. HI titers of target strains from CDC report

Strain name	CA04(F859)	BR07(F968)	BR07(F998)
CA04(F859)	1280	160	160
BR07(F968)	1280	640	640
BR07(F998)	80	160	80

*Figure-Response 1 & Table-Response1 are derived from CDC report: INFORMATION FOR THE VACCINES AND RELATED BIOLOGICAL PRODUCTS ADVISORY COMMITTEE, CBER, FDA February 21, 2008, Page 26. (<http://www.fda.gov/default.htm>)

For case of TX12 (A/Texas/50/2012) to SWZ13 (A/Switzerland/9715293/2013), we obtained data from CDC report again (Figure-Response 2 & Table-Response 2), as being shown below. Similarly, we transformed the experimental data to Dab distance and found the highly contradictory conclusion as being shown in Table-Response 3. That’s why we didn’t use this case to evaluate CE-BLAST.

HEMAGGLUTINATION INHIBITION REACTIONS OF INFLUENZA H3 VIRUSES WITH 20nM OSELTAMIVIR, 4 HA UNITS/ 50 MICROLITERS (2015/01/27)

		REFERENCE FERRET ANTISERA						HA	DATE	
		3C.1		3C.2a		3C.3a				
		EGG	MDCK	SIAT	SIAT	EGG	SIAT	GROUP	DATE	PASS
REFERENCE ANTIGENS		TX/50	TX/50	TX/50	MI/15	SZ/13	SZ/13		COLL	
1	A/TEXAS/50/2012	2560	1280	1280	640	320	160	3C.1	2012/04/15	E5
2	A/TEXAS/50/2012	2560	2560	5120	1280	640	640	3C.1	2012/04/15	M1/C2
3	A/TEXAS/50/2012	640	1280	1280	640	160	320	3C.1	2012/04/15	M1/C152
4	A/MICHIGAN/15/2014	160	320	320	1280	80	160	3C.2a	2014/09/24	M1/S1
5	A/SWITZERLAND/9715293/2013	320	80	80	320	320	80	3C.3a	2013/12/06	E4/E2
6	A/SWITZERLAND/9715293/2013	80	80	80	160	160	320	3C.3a	2013/12/06	S1S2/S2
7	A/NORTH CAROLINA/13/2014	320	80	80	320	320	80	3C.3a	2014/04/15	E5
8	A/NORTH CAROLINA/13/2014	80	80	160	160	160	320	3C.3a	2014/04/15	S2

Figure-Response 2*. screenshots from CDC report

Table-Response 2*. HI titers of target strains from CDC report

Strain name	TX/50(EGG)	TX/50(MDCK)	TX/50(SIAT)	SZ/13(EGG)	SZ/13(SIAT)
TX/50(EGG)	2560	1280	1280	320	160
TX/50(MDCK)	2560	2560	5120	640	640
TX/50(SIAT)	640	1280	1280	160	320
SZ/13(EGG)	320	80	80	320	80
SZ/13(SIAT)	80	80	80	160	320

*Figure-Response 2 & Table-Response 2 are derived from CDC report: INFORMATION FOR THE VACCINE AND RELATED BIOLOGICAL PRODUCTS ADVISORY COMMITTEE CBER, FDA Global Surveillance and Virus Characterization March 4, 2015 CDC, page 27. (<http://www.fda.gov/default.htm>)

Table-Response 3. Experimental antigenic distance of target strains

Stain pairs	Experimental Dab	Antigenic similar or drift
TX/50(EGG) VS SZ/13(EGG)	2.828	similar
TX/50(EGG) VS SZ/13(SIAT)	8	drift
TX/50(MDCK) VS SZ/13(EGG)	4	similar
TX/50(MDCK) VS SZ/13(SIAT)	4	similar
TX/50(SIAT) VS SZ/13(EGG)	5.65	drift
TX/50(SIAT) VS SZ/13(SIAT)	4	similar

However, the drifted strains of your concern are highly important. We checked our paired distance of all strains and make the results into a distance profile according a time window of 10 years with five-year overlapping, as being shown in Figure-Response 3.1-3.9. For each sub-graph, antigenic distances between all paired strains within 10 years were plotted into a map by a classical method of ordinal multidimensional scaling (MDS)¹. Those strains far away from the main cluster would be normally viewed as the drifted strains.

Taking Figure-Response 3.5 as an example, it is easy to notice that A/Netherlands/35/93 and A/Netherlands/5/93 drifted far away from the main cluster, where these two strains were reported as reassortant influenza viruses from pigs². Similarly, the other sub-graphs have been attached here.

Figure-Response 3.5. Antigenic clustering of influenza A (H3N2) viruses from 1988 to 1998.

Figure-Response 3.1. Antigenic clustering of influenza A (H3N2) viruses from 1968 to 1978.

Those antigenic drifted strains with experimental evidences were marked with red circle in each sub-graph. It can be found that strains with significant antigenic drift from others can be detected such as A/Pavia/07/2014³, A/Kansas/13/2009⁴, A/Ontario/1252/2007⁵, A/Hong Kong/1774/99⁶, A/Netherlands/35/93²,

A/Netherlands/5/93², A/Scotland/160/1993⁷, A/Philippines/2/1982⁸,
A/Victoria/112/1976⁹, A/Tokyo/1/1975¹⁰, A/Victoria/3/1975¹⁰.

Figure-Response3.2. Antigenic clustering of influenza A (H3N2) viruses from 1973 to 1983.

Figure-Response 3.3. Antigenic clustering of influenza A (H3N2) viruses from 1978 to 1988.

Figure-Response 3.7. Antigenic clustering of influenza A (H3N2) viruses from 1998 to 2008.

Figure-Response 3.8. Antigenic clustering of influenza A (H3N2) viruses from 2003 to 2013.

Figure-Response 3.9. Antigenic clustering of influenza A (H3N2) viruses from 2008 to 2014.

1. Kruskal, J. B, et al. Multidimensional scaling by optimizing goodness of fit to a nonmetric hypothesis. *Psychometrika*. 29 (1): 1-27
2. Claas EC, et al. Infection of children with avian-human reassortant influenza virus from pigs in Europe. *Virology*. 1994 Oct; 204(1):453-7.
3. Piralla A, et al. Swine Influenza A (H3N2) Virus Infection in Immuno-compromised Man, Italy, 2014. *Emerg Infect Dis*. 2015 Jul;21(7):1189-91
4. Cox CM, et al. Swine influenza virus A (H3N2) infection in human, Kansas, USA, 2009E merging *Infect. Dis*. 17 (6), 1143-1144 (2011)
5. Bastien N, et al. Parotitis in a child infected with triple-reassortant influenza A virus in Canada in 2007. *J Clin Microbiol* 2009 Jun;47(6):1896-8
6. Gregory V, et al. Infection of a child in Hong Kong by an influenza A H3N2 virus closely related to viruses circulating in European pigs. *J Gen Virol*. 2001 Jun; 82(Pt 6):1397-406.
7. Ellis JS, et al. Genetic and antigenic variation in the haemagglutinin of recently circulating human influenza A (H3N2) viruses in the United Kingdom. *Arch Virol*. 1995; 140(11): 1889-904.
8. Arroyo JC, et al. Influenza A/Philippines/2/82 outbreak in a nursing home: limitations of influenza vaccination in the aged. *Am J Infect Control*. 1984 Dec; 12(6): 329-34.
9. Lamnikova SS, et al. Study of the antigenic specificity of hemagglutinin of influenza viruses type A by quantitative radioimmunoassay. Comparative study of differences in the specificity of H3 hemagglutinin in epidemiologically active strains. *Vopr Virusol*. 1978 Jan-Feb; (1): 15-9.
10. Rovnova ZI, et al. Changes in antigenic determinant H3 in type A influenza virus. *Vopr Virusol*. 1978 May-Jun; (3): 282-6.

We have added part of above into *RESULTS* section at page 14 as follows:

Antigenic drift events were also scanned by checking the predicted antigenic distance between strains pairs. The predicted results were mapped into distance profiles according a time window of 10 years with five-year overlapping, by a classical method of ordinal multidimensional scaling (MDS)¹. Those strains far away from the main cluster would be normally viewed as the drifted strains, as being shown in supplementary Figure 7.1-7.9. It can be found that those antigenic drifted strains with experimental evidences can be detected, such as A/Pavia/07/2014³, A/Kansas/13/2009⁴, A/Ontario/1252/2007⁵, A/Hong Kong/1774/99⁶, A/Netherlands/35/93², A/Netherlands/5/93², A/Scotland/160/1993⁷, A/Philippines/2/1982⁸, A/Victoria/112/1976⁹, A/Tokyo/1/1975¹⁰, A/Victoria/3/1975¹⁰.

Third, it is not clear how the antigenic distances are associated with those in the serological assays and it is very difficult to believe the useful of this method.

Maybe we didn't make it clear to biologists. The value of CE-BALST score is NOT directly associated with the serological values. In sequence BLAST tool¹¹, it has been widely accepted that the comparative level of similarity/identity score can indicate gene functional similarity. The score in CE-BLAST plays pretty much the same role as that in sequence BLAST, where the score here is designed to infer antigenic similarity. I hope this can help you understand better.

11. Altschul, S.F. et al (1990) "Basic local alignment search tool." J. Mol. Biol. 215:403-410.

2. The data used in the analyses had too many noises. In the tables listed in supplementary file, the HI data were processed with different protocols. For example, some HI data were processed using turkey red blood cells and some using guinea pig red blood cells. Some HI data were processed with the addition of neuraminidase inhibitors. The authors can not simply use mean titers for their analyses. Overall, current analyses are still too crude to be believed.

In terms of titer values, HI data is indeed noisy after being processed with different protocols and different platforms. But when considering the **classification of antigenic similar or drift**, actually the indication is not that noisy as the titer value looks. For instance, after we transformed the titer values into "similar" or "drift" according to experimental Dab cutoff of 4, only 24 strain pairs of 751 pairs showed contradicting results in our 3867 dataset. Please not that, we take the classification results to validate our model, instead of the simple titer values. So averaging the titer values only affected the classification results of those 24 pairs.

Your concern is reasonable. We deleted those 24 noisy pairs to clean our dataset. The AUC value can rise from 0.917 to 0.923, as below:

Figure-Response 4. AUC results of CE-BLAST after deleting 24 noisy pairs

3. The blast result to influenza database showed that ConH3 sequence is 100% identical to the HA protein sequence of A/Kentucky/01/2007(H3N2). It would be difficult to image this field strain can generate a broad protective spectrum. The authors may need to understand an influenza virus can cross react with those viruses which are circulated before or after this virus. For example, the H3N2 viruses in 2007 could have cross activities (to different extents) with those viruses in 2004 and 2009. The results showed that the sera from ConH3 had a titer of 12,150 against WI05, 2786 against BR07, 2111 against FJ02, and not for MO99. Thus, this result suggested that the ConH3 is most likely designed from a strain and would be expected to have similar immunity to this strain. In summary, the broad protective spectrum for ConH3 is not convincing.

The case of “Con H3” is used to illustrate the usefulness of CE-BLAST in predicting antigenicity, rather than finding the most broad-spectrum vaccine from historical data. Con H3 sequence is a consensus pattern derived from sequences of 2006-2009, while consensus is derived through multiple sequence alignment of a group of sequences including A/Kentucky/01/2007. It’s possible to see a consensus is identical to one of representative sequences within the group. However, this is unlikely to influence the prediction results with our model that predicts the antigenic distance by topologic similarity among different strains. The reviewer raised an interesting alternative possibility to test our model, unfortunately, no binding data was published for A/Kentucky/01/2007. Additionally, our predicted protective spectrum from Figure 3 showed that “Con H3” cover over 30% of 681 historical strains, mostly from year 2002 to year 2010, which may be further tested in the future experiments.

Reviewer #2 (Remarks to the Author):

I have studied the revised version of the manuscript, with particular emphasis on the reasonable measurement of antigenicity similarity and features' contribution in the model. Given the renewed descriptions and results, I am now convinced of the

reasonability of their methodology. I do not have any more comments.

Thanks.

Reviewer #3 (Remarks to the Author):

I think you have done a good job of addressing my original concerns.

Great Thanks.

Additionally, the validation using only influenza invites comparisons against existing influenza-based methods and this reviewer is not convinced this represents an advance on what we already have. To demonstrate generality of the method this reviewer recommends validation using a second pathogen.

Thanks for your suggestion. We added a new paragraph “Prediction on dengue virus confirmed by experimental antisera results” in the *RESULTS* section on page 14:

Prediction on dengue virus confirmed by experimental antisera results

To test the generality of our method, we searched the literature for a second pathogen with complete PDB structures of antigens, well-defined epitope areas, and large-scale experimental data (serum titer or mono-Ab binding). In the end, the antisera data of dengue virus (DENV) from a large-scale study on African green monkey were collected from Leah’s lab¹² to validate our algorithm. In Leah’s experiments, titers were provided for 36 sera samples derived from monkeys injected with corresponding vaccine strains against 47 tested strains of four different DENV serotypes. After removing uninterpretable data, including undone and self-reactive titers <10, the remained data of 1072 strain pairs were included as our validation set. 47 envelope (E) sequences were retrieved according to Genbank ID provided by the paper¹² and 7 antigenic sites were collected from Pitcher’ report¹³. Finally, structures were modeled for CE-BLAST analysis by following the guideline of our method.

It was noted that titer value in influenza case can be transformed into D_{ab} , and D_{ab} above 4 is widely accepted as a threshold of antigenic drift. But no such threshold can be retrieved in DENV case. According to experimental results¹², over 90% of self-reactive titer value is over 20. Under such circumstances, three different values of 15, 20 and 40 were tentatively chosen as classification thresholds for further testing. Accordingly, the classification results of CE-BLAST achieved AUC value of 0.857, 0.894 and 0.899 respectively on the 1072 data (Figure-Response5).

Figure-Response5. ROC curve of 1072 DENV pairs under different thresholds.

Secondly, the experimental antigenicity between DENV strains was mapped into Figure 5A-B by MDS method¹ after data cleaning and normalization (See *ONLINE METHOD* section). The overall antigenicity distance showed that serotype 1 is the closest to serotype 4 whereas serotype 2 is the furthest from the others. The topologies of antigenic clustering between those 47 strains were next examined and illustrated in Figure 5C according to CE-BLAST scores. The four serotypes of DENV can be predicted and clustered respectively by CE-BLAST. Meanwhile, our results showed that serotype 1 was firstly clustered with serotype 4, and then followed by serotype 3 and 2. This fully matched with the antigenic mapping of experimental titers accomplished by MDS¹ (Figure 5A-B).

Within each individual serotype group, CE-BLAST results also agree well with experimental results. Taking serotype 1 as an example, strain DENV1/Vietnam/2008-BID-V1937 (DENV1-V1937) in Figure 5C was firstly clustered with DENV1/Thailand/1964/16007 (DENV1-T16007), and then with DENV1/Myanmar/2005/61117 (DENV1-M61117), followed by others. This topological structure indicates that the antigenicity of DENV1-V1937 is the closest to that of DENV1-T16007 and secondly to that of DENV1-M61117. Interestingly, Leah's results from either one-month or three-month post-infection sera confirmed our prediction that the titer values of DENV1-V1937 vs. DENV1-T16007 is the highest among all pairs of DENV1-V1937 vs. 47 tested strains, and followed by DENV1-M61117 with secondly high titer values¹², indicating the close antigenicity between DENV1-V1937 and DENV1-T16007.

We further did strain clustering according to sequence similarity or structure similarity of E antigens. In sequence-based clustering, serotype 1 was firstly clustered with serotype 3, and then followed by serotype 2 and 4 in Figure 5D, which disagree with Leah's experimental results. Importantly, the similarity between DENV1-V1937 and DENV1-T16007 cannot be inferred at all from the sequence-based method. In structure-based method, clustering tree was constructed based on RMSD of structure comparison¹⁴, with results displayed as Figure 5E. Similar to sequence clustering, structure clustering can suggest neither inter- serotype relationship nor the relative strain similarity within a DENV serotype. Therefore, CE-BLAST model seems to give the best inference of serological similarity comparing to sequence- or structure-based methods.

Figure 5. Performance of CE-BLAST on dengue virus. (A) 3-D antigenic mapping of 28 dengue virus strains based on experimental data by MDS. (B) 2-D antigenic mapping of 28 dengue virus strains based on experimental data by MDS. (C) Antigenic clustering based on CE-BLAST score. (D) Phylogenetic tree of E protein sequence. (E) Clustering tree based on RMSD score of Multiprot¹⁴.

Also, we added a new paragraph in *ONLINE METHOD* section in page 26:

CE-BLAST based antigenic mapping of Dengue virus E protein

To generate the complete data for antigenic mapping, 19 tested strains with undone (empty value) in each line were removed from table S3 of Leah's study¹². For those with antisera value labeled as <10, we arbitrarily set a value of 5 simply for calculation. 28 tested strains were remained with 36 antisera values for each. Then for each line of tested strain, titer values were normalized within 0 to 1 by setting the highest antisera value as 1 (Supplementary Table 9). Finally, the antigenic mapping was done by ordinal multidimensional scaling (MDS)¹ according to normalized titers.

References:

12. Leah C. Katzelnick, et al. Dengue viruses cluster antigenically but not as discrete serotypes. *Science*. Vol 349 Issue 6254, 1338-1342.
13. Trevor J. Pitcher, et al. Functional analysis of dengue virus (DENV) type 2

envelope protein domain 3 type-specific and DENV complex-reactive critical epitope residues. *Journal of General Virology* (2015), 96, 288-293.

14. Shatsky, M., et al. A method for simultaneous alignment of multiple protein structures. *Proteins* 56, 143-156.

Reviewers' comments:

Reviewer #1 (Remarks to the Author):

In the revised version, the authors listed some examples but have not fundamentally addressed my concerns about the accuracy and usefulness of the proposed tool.

1. Again, the fundamental problem for this tool is still that it is impossible to predict structural changes caused by one or a small number of amino acids with current state-of-art structural predicting algorithms. Thus, the results using their modeling would not be accurate to model accurately antigenic variations in influenza A viruses. Figure 4 is an example with missed key H3 variants. For example, the re-emerging H3N2v is antigenically similar to those viruses in the middle of 1990s. H3N2v has caused >2000 human cases but were not identified from this study. I have not seen values for the current tool.

2. It is incorrect to simply average HI data from various sources. The authors understood the challenge but did not fundamentally address those issues I raised before about the data issues. The authors must understand antigenic difference are much more complicated than simply comparing the serological titers between two viruses.

3. The authors still stated "Design of Potential Broad-Spectrum Vaccine" about conH3. The fundamental issues I raised before were not addressed at all.

Reviewer #4 (Remarks to the Author):

The authors propose a new structure- and physicochemical-property-based matching tool analogous to BLAST to identify antigenically important changes in epitopes. This is an interesting idea, which would be valuable for fast antigenic matching from sequence data where no appropriate serology exists. I was asked to comment on earlier reviews of the manuscript, in particular on whether the authors have convincingly demonstrated the utility of the method.

Regrettably I don't find that they have demonstrated this sufficiently clearly. The technique itself may well be extremely effective, and the primary dataset used is a very sensible one, comprising a wide range of influenza A(H3N2) sequence and hemagglutination inhibition data, which is ideal for this kind of work, but I find the validation and comparison with other tools less than ideal.

First, the authors deliberately do not compare CE-BLAST to existing tools that rely on assay data to train the model, as they believe this would be an unfair comparison for their tool, which is not tuned using the assay data. However, I suspect it would also show the tool to be less effective than these other techniques for the same ultimate goal.

Furthermore, some tuning was presumably done based on the assay data they had available to them. At the very least how was 10 chosen for beta in equation 1 on p.23 of the revised manuscript? And were the equations themselves that those parameters affect chosen from a range of different possibilities, and if so, how?

In response to reviewer 1, the authors also point out that the CE-BLAST scores do not directly correspond to serological values. This is completely understandable for this kind of tool, especially when they have chosen not to train it using assay data, but it also detracts from the utility of the

tool. For instance, a tool which just looked at the date differences of isolates being compared, and assessed antigenic similarity based on the number of years separating the samples might do quite well if it did not need to offer a "direct association" with serology due to the strong antigenic drift observed in H3N2 (though I'm not suggesting that anything that crude would do as well as this tool).

The comparison with assay-trained tools would therefore have been valuable to understand what the state of the art is in that area - it wouldn't have been necessary to show this tool was better since it has a different target in mind, but a comparison with structural alignment tools, while valuable, doesn't tell the whole story when none of them were really designed for this task as I understand it.

The validation against Con H3 / A/Kentucky/01/2007 would also have been much more valuable if it had instead identified something unusual - a similarity based on structure or physico-chemical properties that was not immediately predictable from the phylogeny, for instance. Even better, if it had incorporated some reverse genetics to demonstrate that the tool could predict a change in antigenic phenotype, ideally a change to make a virus more antigenically similar to some phylogenetically distant viruses, while moving away from other, closer viruses resulting from some readily identifiable structural change.

None of this necessarily detracts from the overall quality of the algorithm designed by the authors, but I do not feel they demonstrate sufficiently clearly how well it works.

Reviewers' comments:

Reviewer #1 (Remarks to the Author):

In the revised version, the authors listed some examples but have not fundamentally addressed my concerns about the accuracy and usefulness of the proposed tool.

1. Again, the fundamental problem for this tool is still that it is impossible to predict structural changes caused by one or a small number of amino acids with current state-of-art structural predicting algorithms. Thus, the results using their modeling would not be accurate to model accurately antigenic variations in influenza A viruses. Figure 4 is an example with missed key H3 variants. For example, the re-emerging H3N2v is antigenically similar to those viruses in the middle of 1990s. H3N2v has caused >2000 human cases but were not identified from this study. I have not seen values for the current tool.

We understand your concern. As a computational model, CE-BLAST may not be able to predict each individual variant. Thus, we test our model based on massive historical data at different dates and compared with those available computational peers assay-trained on influenza A virus H3N2, as being newly added in Fig 2. The results showed that our assay-independent model perform similarly well with those assay-trained tools. More importantly, this unsupervised method would be more valuable for new pathogens where no appropriate serology exists, which was further demonstrated on inter-subtypic Dengue virus and cross-virus case between Zika and Dengue.

Figure 2 | Performance comparison between CE-BLAST and peers on mutual HI data of 3867 HA pairs of influenza H3 from 1968 to 2013. The X axis represents different size of training data (represented by spanning years) at different testing dates with an increasing window of 5 years. Blue bars show the size of training data (1968-year X), whereas grey bars represents the size of remaining data as testing (year X-2013), with the corresponding values indicated on the right. Each coloured line shows the

performance (AUC value) of the computational models, corresponding to the value on the left.

2. It is incorrect to simply average HI data from various sources. The authors understood the challenge but did not fundamentally address those issues I raised before about the data issues. The authors must understand antigenic difference are much more complicated than simply comparing the serological titers between two viruses.

In this version, we gave up averaging HI values and used the classical cutoff (Dab of 4) to judge the antigenic similar or drifted pairs according to experimental results. Only pairs of consistent classification from experimental results are taken in this version as testing dataset.

3. The authors still stated "Design of Potential Broad-Spectrum Vaccine" about conH3. The fundamental issues I raised before were not addressed at all.

In this re-constructed version, we gave up the statement, and only took "Con H3" as a new validation case. Thus we have changed the statement to "Reliable prediction for a new influenza H3 vaccine".

Reviewer #4 (Remarks to the Author):

The authors propose a new structure- and physicochemical-property-based matching tool analogous to BLAST to identify antigenically important changes in epitopes. This is an interesting idea, which would be valuable for fast antigenic matching from sequence data where no appropriate serology exists. I was asked to comment on earlier reviews of the manuscript, in particular on whether the authors have convincingly demonstrated the utility of the method.

Regrettably I don't find that they have demonstrated this sufficiently clearly. The technique itself may well be extremely effective, and the primary dataset used is a very sensible one, comprising a wide range of influenza A(H3N2) sequence and hemagglutination inhibition data, which is ideal for this kind of work, but I find the validation and comparison with other tools less than ideal.

First, the authors deliberately do not compare CE-BLAST to existing tools that rely on assay data to train the model, as they believe this would be an unfair comparison for their tool, which is not tuned using the assay data. However, I suspect it would also show the tool to be less effective than these other techniques for the same ultimate goal.

Great thanks for your constructive suggestion. In this version, we made complete comparison to accessible tools on influenza and re-constructed the manuscript:

1) Change the title of the manuscript to "CE-BLAST: Making it possible to precisely compute antigenic similarity for newly emerging pathogens"

2) We made comprehensive comparison between our model and the existing assay-trained tools in a way that the predictions were simulated by different training dataset from 1968 to 2013 via a sliding window of 5 years. The results were summarized in Figure 2, and demonstrated that our method performs similarly well with peers at current testing date. But in the early years, the performance of assay-trained models varied differently as being expected.

In response to your concern, on paragraph has been added into the paper as below:

In addition, we simulated the prediction results of CE-BLAST and other peers by different training data size with different date from 1968 to 2013 via a sliding window of 5 years, with training data continually increasing and testing continually decreasing (*Online Methods*). As almost all the available *in-silico* tools on influenza comprise supervised models, three assay-trained methods were chosen as representative peers including Lees' method¹⁷, AntigenCO¹⁵, and a most recent method from Qiu¹⁶, considering their repeatability and accessibility. As shown in **Fig. 2**, the overall prediction abilities of supervised models varied differently across different dates, with AUC value below 0.65 at the beginning of testing period in 1972. When the size of training data kept increasing, their performances become relatively stable with AUC value over 0.8 after 1992. In comparison, CE-BLAST gave high performance of AUC value around 0.9 from the beginning in 1972, and maintained a consistently high and stable AUC value across the whole testing periods. As an unsupervised method, the prominent value of CE-BLAST appears to be able to process fast antigenic matching of new antigens where no appropriate serology exists. Subsequently, we thus extended the prediction ability of our tool to new antigens from different pathogens.

Figure 2 | Performance comparison between CE-BLAST and peers on mutual HI data of 3867 HA pairs of influenza H3 from 1968 to 2013. The X axis represents different size of training data (represented by spanning years) at different testing dates with an increasing window of 5 years. Blue bars show the size of training data (1968-year X), whereas grey bars represents the size of remaining data as testing (year X-2013), with the corresponding values indicated on the right. Each coloured line shows the performance (AUC value) of the computational models, corresponding to the value on the left.

Furthermore, some tuning was presumably done based on the assay data they had available to them. At the very least how was 10 chosen for beta in equation 1 on p.23 of the revised manuscript? And were the equations themselves that those parameters affect chosen from a range of different possibilities, and if so, how?

Thank you for pointing this out. We have analyzed the distribution of each parameters in equation 1, and found the distribution of environmental similarity score for each residues is from 0 to 1, while the other two parameters are natural number. To balance the order of magnitude for different parameters, the beta was simply set as 10 here, so that the order of parameter magnitude reached the same level.

In response to reviewer 1, the authors also point out that the CE-BLAST scores do not directly correspond to serological values. This is completely understandable for this kind of tool, especially when they have chosen not to train it using assay data, but it also detracts from the utility of the tool. For instance, a tool which just looked at the date differences of isolates being compared, and assessed antigenic similarity based on the number of years separating the samples might do quite well if it did not need to offer a "direct association" with serology due to the strong antigenic drift observed in H3N2 (though I'm not suggesting that anything that crude would do as well as this tool).

Maybe we didn't make it clear. Thank you for pointing this out.

Although CE-BLAST scores do not directly correspond to serological values, the scores of different strain pairs can be compared relatively. Furthermore, the classification of antigenic similar or drift can be suggested by the empirical thresholds recommended by our model. Most importantly, as you said, our model would be valuable for new pathogens where no appropriate serology exists at all.

The comparison with assay-trained tools would therefore have been valuable to understand what the state of the art is in that area - it wouldn't have been necessary to show this tool was better since it has a different target in mind, but a comparison with structural alignment tools, while valuable, doesn't tell the whole story when none of them were really designed for this task as I understand it.

We totally agree with you on this point.

In this version, we compared CE-BLAST with available assay-trained tools and found comparatively good performance on influenza. In addition, we highlighted the utility of our model for new pathogens, thanks to your kind guidance. This utility was then proved on inter-subtypic dengue virus and new pathogens of zika virus.

The validation against Con H3 / A/Kentucky/01/2007 would also have been much more valuable if it had instead identified something unusual - a similarity based on structure or physico-chemical properties that was not immediately predictable from the phylogeny, for instance. Even better, if it had incorporated some reverse genetics to demonstrate that the tool could predict a change in antigenic phenotype, ideally a change to make a virus more antigenically similar to some phylogenetically distant viruses, while moving away from other, closer viruses resulting from some readily identifiable structural change.

Your comments are highly enlightening!

In this reconstructed version, we organized the validation of our method at three different levels: 1) intra-subtypic on influenza A virus H3N2; 2) inter-subtypic on dengue virus; and 3) cross-virus cases between dengue and zika virus.

Your expectation is fully demonstrated in two new parts added in this version:

- 1) For inter-subtypic case of dengue virus, those phylogenetically distant dengue viruses can be successfully detected as antigenically similar pairs, while neither the traditional sequence-based or structure-based comparison method can infer this.
- 2) For cross-virus cases, not only the antigenically similar pairs was suggested, but also the cross-reactive epitope between dengue and zika virus was precisely delineated via CE-BLAST.

For further details, kindly refer to the two new validation parts in RESULTS and DISCUSSION as being attached below:

Correct prediction of serological topology for DENV subtypes

To further test the generality of CE-BLAST toward newly emerged pathogens, the antisera data of DENV were collected from a large-scale study on the African green monkey⁶. In this cited study, 36 sera samples derived from monkeys injected with corresponding vaccine strains were tested individually against 47 DENV strains of four different serotypes. After removing un-interpretable data with undone and self-reactive titers <10, the remained titer data of 1072 strain pairs were included as our validation set. We modeled 47 E protein structures for CE-BLAST according to the sequences provided in the paper⁶. Unlike for influenza, no empirical titer threshold has been reported as being able to classify antigenic similarity or variance for DENV cases. According to the statistics on available data⁶, the titer value of over 90% of self-reactive pairs was over 20. Thus, three different values of 15, 20, and 40 were tentatively chosen as classification thresholds for further testing. Accordingly, the classification results of CE-BLAST achieved AUC values of 0.857, 0.894, and 0.899, respectively, on the 1072 strain pairs.

Next, the antigenic grouping results of CE-BLAST were compared with that from classical sequence similarity and structure similarity based on experimental serological topology. **Fig. 4a-b** show the serological topology of experimental grouping between DENV strains by the multidimensional scaling (MDS) method¹⁸ after data cleaning and normalization (*Online Method*). It could be observed that serotype 1 clusters closely with serotype 4 whereas serotype 2 clusters the furthest from the remainder. In **Fig. 4c** of the CE-BLAST results, the four serotypes of DENV could be correctly predicted and clustered. In comparison to grouping topology, serotype 1 was first clustered with serotype 4, whereas serotype 2 is the furthest from the remaining strains, which fully matches with the experimental topology (**Fig.4a-b**). However, in sequence-based clustering of **Fig. 4d**, serotype 1 was first clustered with serotype 3, followed by serotype 2 and last with serotype 4, which disagrees with the experimental results. Neither could the structure method achieve the correct topology for DENV subtypes, as displayed in **Fig. 4e**.

In addition to inter-subtypic DENV, CE-BLAST also yields better prediction for intra-subtypic pathogens than sequence-based or structure-based methods. Taking serotype 1 as an example, strain DENV1/Vietnam/2008-BID-V1937 (DENV1-V1937, marked with a star) in **Fig. 4c** was first clustered with DENV1/Thailand/1964/16007 (DENV1-T16007, marked with a dot), and then with DENV1/Myanmar/2005/61117 (DENV-M61117, marked with a cross), followed by others.

This topological structure indicated that the antigenicity of DENV1-V1937 was closest to that of DENV1-T16007 and secondly to that of DENV1-M61117. Our prediction is strongly corroborated by experimental results from either one-month or three-month post-infection sera. The titer value of DENV1-V1937 vs. DENV1-T16007 is the highest among all pairs between DENV1-V1937 and the 47 tested strains, followed by DENV1-M61117 with the second highest titer values, indicating the close antigenicity between DENV1-V1937 and DENV1-T16007. In contrast, neither sequence-based nor structure-based methods could suggest the best serological relationship within DENV subtypes, as being displayed in **Fig. 4d-e**. Therefore, the CE-BLAST model appears to give the best inference of serological similarity for DENV subtypes compared to the classical sequence- or structure-based methods.

Figure 4 | Subtype grouping of dengue virus by CE-BLAST. (a) 3-D antigenic mapping of 28 dengue virus strains based on serological data from Katzelnick et al⁶ by MDS. (b) 2-D antigenic mapping of Fig. 4a. (c) Antigenic clustering of 47 strains by CE-BLAST similarity score. (d) Traditional grouping by sequence phylogenetic tree of 47 E protein sequences. (e) Traditional grouping by structural clustering tree of 47 E proteins based on RMSD scores of Multiprot¹⁹.

Successful identification of cross-reactive epitopes between DENV and ZIKV

The obtained results indicated the unique ability of CE-BLAST to predict antigenic similarity for intra- and inter- subtypes of new pathogens. Next, CE-BLAST was tested across different viruses to detect potential cross-reactivity between the latest arising pathogen of ZIKV and the available pathogen of DENV. ZIKV is a member of the Flavivirus family, which recently emerged from Brazil and quickly became a significant public health concern. Evidence shows that ZIKV infections may lead to neurological complications such as Guillain–Barré syndrome in adults²⁰ and microcephaly in newborns¹. Several reports discovered that the antibodies isolated from patients with dengue had the potential to cross-react with ZIKV^{21,22}. As the main target of neutralizing antibodies, the E protein was reported to share high structural similarity with a root-mean-square deviation (RMSD) of 1.1 Å and overall sequence identity of 53.9% between ZIKV and DENV²³.

To predict the potential cross-reactive epitopes between ZIKV and different DENV subtypes, four representative E proteins were randomly sampled from our dataset for each DENV subtype as well as for ZIKV. For the convenience of computer screening, round patches were collected for each residue on the protein surface after structure modeling of E monomers and dimers, respectively (*Online Methods*). Then, the surface patches of ZIKV were compared with the corresponding patches in DENV subtypes through CE-BLAST. Potential cross-reactive patches (CRPs) were marked when their similarity scores rose above a certain threshold. The frequency of CRP labelled by the center residue was mapped onto a heat map resulting from binary comparison between 4 ZIKV and 4 DENV structures, as shown in **Fig. 5a**, with different rows representing DENV subtypes. Additional results can be found in **Supplementary Fig. 3** and **4**. Although the *in-silico* cross-reactivity frequency could vary among different subtypes of DENV, strongly consistent CRPs in domain II and additionally weak CRPs in domain I could be detected across DENV subtypes (**Fig. 5a3**). Our prediction is supported by the experimental results from a study by Stettler et al on testing the reactivity ability of domain I/II and domain III in the E protein monomer²².

It is noted that the true epitope is often irregularly shaped, whereas the above circular surface patch is artificially over-simplified for convenient screening purposes. The same CRE residue may be contained by different artificial patches and the particular artificial patches covering sufficient CRE residues are more likely to constitute CRPs. Thus, the overlapping of such CRPs likely

indicates the location of true epitopes. Subsequently, individual residues in each *in-silico* CRP of a given subtype are first mapped onto the 3-D surface of the E antigen. Concentrated areas above the average are shown in green in **Fig. 5b1-4**, hinting at the potential cross-reactive epitope (CRE) between ZIKA and DENV subtypes, respectively. Subsequent overlapping of subtypic CREs suggested the green region in **Fig. 5b5** as the potential CRE across DENV subtypes to ZIKV virus.

A similar strategy was applied to E protein dimer structures. The computed CRE across DENV subtypes was strongly hinted in domain II, albeit only slightly in domain I and III from the opposite chain in the dimer structure (**Supplementary Fig. 5**). The whole CRE predicted for the E dimer involves 14 surface residues, as labelled in **Fig. 5c1**. Notably, the computed CRE is highly overlapping with results from crystallization work of immune-complexes by Barba-Spaeth et al²¹. In particular, 71% of our computed CRA residues are located in the binding interface derived from the structure complex between the antibody and the E protein dimer (**Fig. 5c2**), with 45% of the important residues suggested by Barba-Spaeth et al included in our prediction (**Fig. 5c3**).

Figure 5 | Predicting the potential cross-reactive epitope between DENV and ZIKV. (a) Workflow of potential cross-reactive area (CRA) detection by CE-BLAST between ZIK and DENV. **a1**: Two E antigens to be compared, with domain I, II, and III marked in yellow, red, and blue, respectively; **a2**: circular patches are screened and compared on the antigen surface; **a3**: the cross-reactive frequency among sampling structures between corresponding patches predicted by CE-BLAST. Each patch is labeled by the center residue in the column, and each row represents four DENV types. Red dashed boxes show the consistent CRAs across different DENV subtypes whereas yellow box shows the weak one. Residues in different domains are marked accordingly on the bars over the heat map. (b) Potential cross-reactive epitope (CRE) mapping to the E monomer structure of ZIKV. **b1-b4**: the predicted CRE is shown in green for four DENV serotypes respectively; **b5**: overlapping CRE of ZIKV across DENV subtypes. (c) Predicted CRE of the E dimer structure of ZIKV, compared with experimental results. **c1**: predicted CRE by CE-BLAST for the E dimer; **c2**: binding interface derived from the crystal structures (PDB id:5LCV); **c3**: important residues computed by interaction force from Barba-Spaeth et al ²². All CREs have been circled for clarity.

None of this necessarily detracts from the overall quality of the algorithm designed by the authors, but I do not feel they demonstrate sufficiently clearly how well it works.

We highly appreciated your insightful comments which have greatly improved the quality of our manuscript!

REVIEWERS' COMMENTS:

Reviewer #1 (Remarks to the Author):

The author said, as a computational model, CE-BLAST may not be able to predict each individual variant. How this model will be useful?

The authors added computational performance for influenza antigenic variants prediction in this revision. The predicting accuracy were up to 90% but earlier ones were less than 40%. The variations of the results were surprising because the authors said their method is a unsupervised method. The descriptions on how the comparisons were performed were too sketchy.

1) What exactly are the data used in the experiments? Do the authors used the same data in previously publications?

2) How the performance were evaluated?

3) The data and results should be provided along with this manuscript. The description on data in the supplementary data do not make sense at all. The authors should provide the exact sequence and serological data used in the studies.

Reviewer #4 (Remarks to the Author):

The authors have gone to great lengths to address the issues raised in my initial review, introducing three new sections to the paper. These substantially change my ability to assess how well the technique works.

They compare CE-BLAST to three published techniques, one of their own and two others, for determining antigenic similarity that use serological data. The comparison paints CE-BLAST in a very favourable light. I might have chosen other more sophisticated techniques, but these would have been significantly harder for the authors to reproduce, and my main concern in my earlier review was to see that this structure/physicochemical-property-based technique was not substantially behind the state-of-the-art for these other classes of model. I am reassured to see that, to the contrary, the technique seems to do as well or better than their selected comparators in this case. AUC for match/mismatch assessment is something that I have not often seen before, with the techniques I am familiar with tending to examine mean absolute error of titre predictions (which itself has drawbacks), but since the technique under review does not directly estimate titre, this seems like a reasonable alternative.

They then demonstrate the ability of the tool to identify antigenically similar viruses, which straightforward phylogenetic, sequence-based and structural alignment tools are unable to recognise. These two examples - between Dengue subtypes and between Dengue and Zika virus - are well outside my area of expertise, but seem to be very positive for the utility of the tool. However, the highlighted comparison between 3 viruses within DENV1 seems somewhat arbitrary, given that we do not know if other viruses are further apart in the CE-BLAST clustering than the serology would suggest.

Nonetheless, and despite the absence of experimental work such as reverse genetics to demonstrate the causal link between predictions and outcome, these results collectively address my concerns about the validity of the tool for identifying antigenically important changes and similarities between epitopes, and in particular as a tool-of-last-resort when no serological data exists, or possibly of first resort before such data is generated.

It's important to note that I remain unfamiliar with any details of structural modelling in general (beyond a professional interest), for instance I still don't understand what a spin-image is, I'm afraid to say, and am unable to comment on the reasonableness or indeed validity of the underlying techniques being proposed. I was also again disappointed that the tool itself is not available online despite this being advertised in the abstract - this is something that must be addressed before publication, along with some mechanism for assuring it will remain accessible, as it would have been much easier to get to grips with this by "playing" with the tool a bit. However, within those (significant) limitations, I am now happy that the manuscript is suitable for publication.

Note that the manuscript is still in need of proof-reading and editing. For instance, it would be good if CE-BLAST itself was spelt correctly throughout(!); I'm not sure the opening sentence of the abstract still connects to the rest of the manuscript as well as it did; a clearer explanation of the spin-image would be extremely useful; Fig 4a seems to add nothing to the manuscript, and Fig 4b seems in a different style to everything else; Fig 3b and 4b-e are almost impossible to read; Fig 4's legend does not explain the symbols on 4c-e, and those symbols do not appear on 4a-b, which would be useful; finally some sentences especially in the discussion don't read very well, describing the technique as unique is unnecessary, and using PTM as an acronym without explanation.

Finally, I think the revised title is inappropriate, or at least the use of the term "precisely". While I am happy that CE-BLAST can compute antigenic similarity, I would reserve "precisely" for the ability to actually estimate the antigenic distance in terms of titre of some other precisely quantifiable measure.

REVIEWERS' COMMENTS:

Reviewer #1 (Remarks to the Author):

The author said, as a computational model, CE-BLAST may not be able to predict each individual variant. How this model will be useful?

Perhaps we didn't describe it clear enough. In the context we actually mean: As a computational model, CE-BLAST cannot guarantee to predict each individual variant correctly.

The authors added computational performance for influenza antigenic variants prediction in this revision. The predicting accuracy were up to 90% but earlier ones were less than 40%. The variations of the results were surprising because the authors said their method is a unsupervised method. The descriptions on how the comparisons were performed were too sketchy.

Maybe we didn't illustrate it clear enough, which lead to your misunderstanding in Figure 2. As an unsupervised method, CE-BLAST maintains a high performance near 0.9 from the beginning (magenta line). In Figure 2, Qiu's method (blue) represents our previously published method¹, not CE-BLAST.

1) What exactly are the data used in the experiments? Do the authors used the same data in previously publications?

The data used in Figure 2 contains mutual HI data of 3867 HA pairs of influenza H3 from year 1968-2013. All data was publicly collected from reports of international organizations and publications as **Supplementary Note 1** described. Artificially

consolidated dataset were listed in **Supplementary Table 9**. The HI dataset from year 2011-2013 was also validated in our previous work¹. In this article, 7 additional testing dates (1973-2013, 1978-2013, 1983-2013, 1988-2013, 1993-2013, 1998-2013, 2003-2013 and 2008-2013) were validated and compared across different computational models.

2) How the performance were evaluated?

The titer results of HI assay can be calculated into the antigenic distance of the two strains (D_{ab}). Two virus strains were defined as antigenic variants experimentally if the $\log^{-1}D_{ab}$ was above 4, otherwise, the pair was treated as antigenic similar. Then, 3867 strain pairs can be classified as antigenic variants or antigenic similar according to their HI test values as golden standards (Described in **Supplementary Note 2**). Then, the results of experimental variant/similar will be compared to that from different computational methods for each strain pair under different thresholds. The ROC (Receiver Operating Characteristic) curve will be created by plotting the true positive rate against the false positive rate at various threshold settings, while AUC value represents the area under the ROC curve. Finally, the AUC values can be obtained to evaluate the overall performance of different computational methods. Note that, AUC value varies between 0 to 1, while 1 represents perfect performance (100% match between golden standard and predicted results).

3) The data and results should be provided along with this manuscript. The description on data in the supplementary data do not make sense at all. The authors should provide the exact sequence and serological data used in the studies.

Good suggestion. The serological data of HI test with corresponding strain names were provided in **Supplementary Table 9** and HA sequence used in this study were summarized in **Sequence data for HA.fasta**. Sequence data used in section *Reliable prediction for a new influenza H3 vaccine* were listed in **Supplementary Table 1**, sequence comparison of Con H3 and reference strains can be found in **Supplementary Figure 2**.

1. Qiu, J. X., Qiu, T. Y., Yang, Y. Y., Wu, D. F. & Cao, Z. W. Incorporating structure context of HA protein to improve antigenicity calculation for influenza virus A/H3N2. *Scientific reports* 6, doi:Artn 31156 10.1038/Srep31156 (2016).

Reviewer #4 (Remarks to the Author):

The authors have gone to great lengths to address the issues raised in my initial review, introducing three new sections to the paper. These substantially change my ability to assess how well the technique works.

They compare CE-BLAST to three published techniques, one of their own and two others, for determining antigenic similarity that use serological data. The comparison paints CE-BLAST in a very favourable light. I might have chosen other more sophisticated techniques, but these would have been significantly harder for the authors to reproduce, and my main concern in my earlier review was to see that this structure/physicochemical-property-based technique was not substantially behind the state-of-the-art for these other classes of model. I am reassured to see that, to the contrary, the technique seems to do as well or better than their selected comparators in this case. AUC for match/mismatch assessment is something that I have not often seen before, with the techniques I am familiar with tending to examine mean absolute error of titre predictions (which itself has drawbacks), but since the technique under review does not directly estimate titre, this seems like a reasonable alternative.

Thank you very much for your comments.

They then demonstrate the ability of the tool to identify antigenically similar viruses, which straightforward phylogenetic, sequence-based and structural alignment tools are unable to recognise. These two examples - between Dengue subtypes and between Dengue and Zika virus - are well outside my area of expertise, but seem to be very positive for the utility of the tool. However, the highlighted comparison between 3 viruses within DENV1 seems somewhat arbitrary, given that we do not know if other viruses are further apart in the CE-BLAST clustering than the serology would suggest.

Reasonable. Here, we trying to display the utility of CE-BLAST in two different levels: 1) model evaluation with 1072 pairs of antisera data, which illustrated the good performance of CE-BLAST on large-scale data; 2) topological evaluations compared with phylogenetic, sequence-based and structural alignment tools. Results showed that CE-BLAST could achieve the correct topology for DENV subtypes (Figure 4c) which is agree well with serological data (Figure 4a and Figure 4b). It is true that other tools may also give comparable prediction to individual strain within DENV1. The 3 viruses arbitrarily selected were only shown as detailed examples within DENV subtypes under the correct topology prediction between DENV subtypes.

Nonetheless, and despite the absence of experimental work such as reverse genetics to demonstrate the causal link between predictions and outcome, these results collectively address my concerns about the validity of the tool for identifying antigenically important changes and similarities between epitopes, and in particular as a tool-of-last-resort when no serological data exists, or possibly of first resort before such data is generated.

Thanks.

It's important to note that I remain unfamiliar with any details of structural modelling in general (beyond a professional interest), for instance I still don't understand what a spin-image is, I'm afraid to say, and am unable to comment on the reasonableness or indeed validity of the underlying techniques being proposed. I was also again disappointed that the tool itself is not available online despite this being advertised in the abstract - this is something that must be addressed before publication, along with some mechanism for assuring it will remain accessible, as it would have been much easier to get to grips with this by "playing" with the tool a bit. However, within those (significant) limitations, I am now happy that the manuscript is suitable for publication.

Sorry for the inconvenience before. Now the web server is fully online. To ensure the full accessibility, we made a mirror site outside of our campus. It can be both accessed at http://badd.tongji.edu.cn/ce_blast/ or http://bidd2.nus.edu.sg/czw/ce_blast/.

Note that the manuscript is still in need of proof-reading and editing. For instance, it would be good if CE-BLAST itself was spelt correctly throughout(!); I'm not sure the opening sentence of the abstract still connects to the rest of the manuscript as well as it did;

Thank you very much for your suggestion. We had proof-reading and editing the whole manuscript as you suggested.

a clearer explanation of the spin-image would be extremely useful;

To clearly explain spin-image, the first paragraph of *Structural fingerprint generation via the "spin-image" system* in **Methods** part was modified as follows:

‘
The spin-image technology was initially designed to solve the 3-D object recognition and reconstruction problems. In this paper, it is used to project the 3-D neighboring environment layout of a target residue to a 2-D array/image through spinning a dynamic plane. One 2-D image will be generated for each epitope residue. Thus, an epitope structure will be described by a collection of 2-D images, defined as spin-images. In this manner, an epitope surface can be represented by a group of spin-images.
,

Fig 4a seems to add nothing to the manuscript, and Fig 4b seems in a different style to everything else;

Fig 4a and Fig 4b were obtained from antisera data by using multidimensional scaling (MDS) techniques. MDS was used here to project points (representing different

strains) from high dimensional space to two or three dimensional space. In that case, Fig 4a is used for better illustration of serological clustering in three dimensional space and Fig 4b contains serological clustering in two dimensional space.

Fig 3b and 4b-e are almost impossible to read; Fig 4's legend does not explain the symbols on 4c-e, and those symbols do not appear on 4a-b, which would be useful;

We uploaded high-resolution figures of Fig 3 and Fig 4 in revised version, and modified the figure legends. Note that Fig 4a and Fig 4b were derived from experimental data after data purification (See *Dengue virus E protein in Application of CE-BLAST to different pathogens*). Several strains illustrated in Fig 4c-e are not contained in Fig 4a-b. In that case, those symbols were not marked in Fig 4a-b.

finally some sentences especially in the discussion don't read very well, describing the technique as unique is unnecessary, and using PTM as an acronym without explanation.

The final sentence was modified as follows:

‘
Subsequent improvements will be further elaborated on Post-translational modification (PTM) antigens, parallel computing, and refined models that are tailor-made for specific proteins.
‘

Finally, I think the revised title is inappropriate, or at least the use of the term "precisely". While I am happy that CE-BLAST can compute antigenic similarity, I would reserve "precisely" for the ability to actually estimate the antigenic distance in terms of titre of some other precisely quantifiable measure.

Thank you for pointing this out, we deleted the word and modified the title as follows:

‘
CE-BLAST: Making it possible to compute antigenic similarity for newly emerging pathogens
‘